# Statistical mechanics of integrable quantum spin systems

Frank Göhmann

Fakultät für Mathematik und Naturwissenschaften
Bergische Universität Wuppertal, Germany

goehmann@uni-wuppertal.de

# Preface

This script is based on the notes the author prepared to give a set of six lectures at the Les Houches School "*Integrability in Atomic and Condensed Matter Physics*" in the summer of 2018. The responsibility for the selection of the material is partially with the organisers, Jean-Sébastien Caux, Nikolai Kitanine, Andreas Klümper and Robert Konik. The school had its focus on the application of integrability based methods to problems in non-equilibrium statistical mechanics. My lectures were meant to complement this subject with background material on the equilibrium statistical mechanics of quantum spin chains from a vertex model perspective. I was asked to provide a minimal introduction to quantum spin systems including notions like the reduced density matrix and correlation functions of local observables. I was further asked to explain the graphical language of vertex models and to introduce the concepts of the Trotter decomposition and the quantum transfer matrix. This was basically the contents of the first four lectures presented at the school. In the remaining two lectures I started filling these notions with life by deriving an integral representation of the free energy per lattice site for the Heisenberg-Ising chain (alias XXZ model) using techniques based on non-linear integral equations.

Up to small corrections the following sections L1-L6 display the lectures almost literally. The only major change is that the example of the XXZ chain has been moved from section L5 to L2. During the school it was not really necessary to introduce the model, since other speakers had explained it before. But for these notes I thought it might be useful to introduce the main example rather early. I also supplemented each lecture with a comment section which contains additional references and material of the type that was discussed informally with the participants.

I am grateful to my colleagues at the University of Wuppertal, Hermann Boos, Michael Karbach and Andreas Klümper, as well as to my long-term collaborators Karol Kozlowski and Junji Suzuki for sharing their considerable insight into the subjects of these lectures. I would like to thank Constantin Babenko and Saskia Faulmann for carefully reading the manuscript and pointing out a number of misprints an inaccurracies in the first version.

# L1   Statistical mechanics of quantum chains

## 1.1   Introduction

Spin systems are the simplest conceivable quantum mechanical systems. In nature the spin occurs in first place as an internal degree of freedom of elementary particles. When many particles bind together in a many-body quantum system like a crystalline solid, the spin may also take the role of a discrete quantum number of collective excitations. In certain experiments on such systems, e.g. on Mott insulators at low temperatures or on ultra-cold atomic gases trapped in optical lattices, it is possible to create 'pure spin excitations'. Such systems are well described, in a certain energy range, by (generalised) Hubbard or Heisenberg models [1], which are in the class of models to be considered in these notes.

Spin systems can be used to illustrate the characteristic properties of quantum mechanics, like its probabilistic nature or the superposition and entanglement of states. For this reason quantum-spin systems are ubiquitous in introductory text books on quantum mechanics [2] and familiar to all graduate students in physics.

- A single spin-$\frac{1}{2}$ is the simplest possible quantum system. Its Hilbert space is $\mathcal{H} = \mathbb{C}^2$ equipped with the Hermitian scalar product.

- Two spins-$\frac{1}{2}$ describe the simplest interacting quantum systems with space of states $\mathcal{H}_2 = \mathbb{C}^2 \otimes \mathbb{C}^2$. A two-spin state may be entangled.

- $N$ many interacting spins-$\frac{1}{2}$ constitute the simplest many-body quantum systems with space of states $\mathcal{H}_N = (\mathbb{C}^2)^{\otimes N}$. Depending on the interaction, these systems may possess complicated highly entangled ground states and may carry collective excitations of various types.

In the thermodynamic (or 'infinite volume') limit, $N \to +\infty$, quantum-spin systems may exhibit critical behaviour. They can be used to study phase transitions and quantum criticality. Apart from the number of constituents $N$, quantum-spin systems typically depend on several interaction parameters in their Hamiltonians. Considering certain scaling limits, in which these parameters depend on $N$ and $N$ is send to infinity, quantum-spin systems may be used to realise quantum field theories on the lattice. Conversely, one may think of quantum-spin systems as of 'fully regularised quantum field theories' [3], meaning that ultra-violet and infra-red regularisations are provided by the fact that the number of spins is finite, and the Hilbert space is regularised due to the fact that spins have a finite number of degrees of freedom.

Arguably, all many-body quantum physics can be phrased in terms of spin systems of sufficiently general type. This should provide enough motivation to thoroughly study their statistical mechanics. It seems to indicate, on the other hand, that the statistical mechanics of quantum-spin systems in general should be too big as a subject for an introductory lecture course. For this reason, after developing part of a general theory, we shall restrict ourselves to integrable quantum-spin systems in these notes. Integrable systems are defined in one spatial dimension (1d) and have a rich algebraic structure underlying, which makes it possible to obtain more or less explicit results, at least for the thermodynamics of some non-trivial quantum spin systems in the infinite-volume limit.

As far as the general theory is concerned we shall introduce and explain what we call the quantum 'transfer matrix approach' to quantum spin systems. In our understanding this approach is a clever and systematic way of attaching an equilibrium statistical operator with any type of local interaction. The equilibrium statistical operator of 1d spin systems will be composed of transfer matrices of a 'classical vertex model' on a two-dimensional

lattice. Such a procedure is non-unique. The non-uniqueness may be utilised to optimise the properties of the transfer matrix for various purposes. e.g. for an efficient calculation of the partition function by quantum Monte Carlo methods [4]. We shall use a construction, introduced by A. Klümper in [5], which is optimised for the use with integrable quantum spin models of Yang-Baxter type. As we shall see, this construction cannot only be used to calculate the free energy per lattice site of such models, but seems to be optimised as well for the calculation of their correlation functions [6].

## 1.2  States and operators

In our mathematical set-up we shall consider systems that are slightly more general than spin-$\frac{1}{2}$ systems in that they have $d \geq 2$ degrees of freedom on a 'local Hilbert space' $\mathcal{H} = \mathbb{C}^d$, equipped with the canonical Hermitian scalar product. We fix a basis

$$\{e_\alpha\}_{\alpha=1}^d \subset \mathbb{C}^d \tag{1.1}$$

in this space.

### 1.2.1  Operators on local Hilbert space

In order to introduce a space of local observables we start with a set of operators $e_\beta^\alpha \in \operatorname{End} \mathbb{C}^d$, $\alpha, \beta = 1, \ldots, d$, defined by their action on the basis (1.1),

$$e_\beta^\alpha e_\gamma = \delta_\gamma^\alpha e_\beta \,. \tag{1.2}$$

Then, for any $A \in \operatorname{End} \mathbb{C}^d$,

$$A e_\gamma = A_\gamma^\beta e_\beta = A_\alpha^\beta \delta_\gamma^\alpha e_\beta = A_\alpha^\beta e_\beta^\alpha e_\gamma \,, \tag{1.3}$$

where (1.2) was used in the last equation. In (1.3) we have also employed the common 'summation convention', implying that Greek indices that occur twice in an expression are summed over from 1 to $d$. We shall keep this convention throughout these notes. Comparing left and right hand sides of (1.3) and taking onto account that the set $\{e_\alpha\}_{\alpha=1}^d$ is a basis, we conclude that

$$A = A_\alpha^\beta e_\beta^\alpha \tag{1.4}$$

and hence that

$$\{e_\beta^\alpha\}_{\alpha,\beta=1}^d \subset \operatorname{End} \mathbb{C}^d \tag{1.5}$$

is a basis of $\operatorname{End} \mathbb{C}^d$. Any basis element $e_\beta^\alpha$ will be called an elementary endomorphism.

The action of a product of two elementary endomorphisms on the basis (1.1) can be computed by means of (1.3),

$$e_\beta^\alpha e_\delta^\gamma e_\varphi = e_\beta^\alpha \delta_\varphi^\gamma e_\delta = \delta_\delta^\alpha \delta_\varphi^\gamma e_\beta = \delta_\delta^\alpha e_\beta^\gamma e_\varphi \,. \tag{1.6}$$

Comparing the left and the right hand side of this equation and using again that $\{e_\alpha\}_{\alpha=1}^d$ is a basis of $\mathbb{C}^d$, we obtain the relation

$$e_\beta^\alpha e_\delta^\gamma = \delta_\delta^\alpha e_\beta^\gamma \tag{1.7}$$

providing a set of structure constants for the algebra $\operatorname{End} \mathbb{C}^d$.

From the first equation (1.3) we see that the identity operator $I_d \in \operatorname{End} \mathbb{C}^d$ is represented by the matrix $A_\alpha^\beta = \delta_\alpha^\beta$. Hence, by (1.4),

$$I_d = e_\alpha^\alpha \,. \tag{1.8}$$

### 1.2.2 Local basis of $L$-site Hilbert space

A multi-spin system is defined on a regular or irregular lattice consisting of $N$ points $\mathbf{x}_k \in \mathbb{R}^n$, $k = 1, \ldots, N$, such that a local Hilbert space $\mathcal{H} = \mathbb{C}^d$ is associated with every point. The Hilbert space of the multi-spin system is then $\mathcal{H}_N = (\mathbb{C}^d)^{\otimes N}$. Since we will soon focus on large integrable lattice systems, we shall assume that $n = 1$ and $\mathbf{x}_k = -L + k$, $k = 1, \ldots, N = 2L$.

We define the embedding of the basis of elementary endomorphisms (1.5) into the lattice,

$$e_{j\,\alpha}^{\ \beta} = I_d^{\otimes(L-1+j)} \otimes e_\alpha^\beta \otimes I_d^{\otimes(L-j)} \in \mathrm{End}(\mathbb{C}^d)^{\otimes 2L} \tag{1.9}$$

for $j = -L + 1, \ldots, L$. With this we can embed the action of '$m$-site operators' into the lattice. For every $A \in \mathrm{End}(\mathbb{C}^d)^{\otimes m}$, $m \leq 2L$, and $\{j_1, \ldots, j_m\} \subset \{-L+1, \ldots, L\}$ we set

$$A_{j_1 \ldots j_m} = A_{\beta_1 \ldots \beta_m}^{\alpha_1 \ldots \alpha_m} e_{j_1\,\alpha_1}^{\ \ \beta_1} \ldots e_{j_m\,\alpha_m}^{\ \ \beta_m} . \tag{1.10}$$

### 1.2.3 Examples

(i) If $A \in \mathrm{End}\,\mathbb{C}^d$, then $A_j$ is a 'single-site' (or ultra-local) operator acting on 'site $j$'.

(ii) Let

$$P = e_\alpha^\beta \otimes e_\beta^\alpha \in \mathrm{End}\,\mathbb{C}^d \otimes \mathbb{C}^d . \tag{1.11}$$

Then

$$P\,\mathbf{x} \otimes \mathbf{y} = e_\alpha^\beta \otimes e_\beta^\alpha \, x^\gamma e_\gamma \otimes y^\delta e_\delta = x^\gamma y^\delta \, e_\alpha^\beta e_\gamma \otimes e_\beta^\alpha e_\delta$$
$$= x^\gamma y^\delta \delta_\gamma^\beta \delta_\delta^\alpha \, e_\alpha \otimes e_\beta = \mathbf{y} \otimes \mathbf{x} . \tag{1.12}$$

Thus, $P$ induces the transposition of factors in a tensor product. In physical terms, it interchanges the states on two sites. This operator, called the transposition or exchange operator, is an important object in the theory of spin systems and occurs in many places.

Most prominently, perhaps, it occurs as 'exchange interaction' in the Heisenberg Hamiltonian

$$H = J \sum_{j=-L+1}^{L} P_{j-1,j} , \tag{1.13}$$

where $J > 0$ and $P_{-L,-L+1} = P_{L,-L+1}$ by definition. This Hamiltonian defines one of the simplest and most generic interacting quantum-spin systems. It is simple in the sense that only neighbouring sites interact and also because $P(A \otimes A) = (A \otimes A)P$ implies that

$$(A \otimes A)P(A^{-1} \otimes A^{-1}) = P \tag{1.14}$$

for all $A \in GL(d)$, the general linear group of invertible endomorphisms on $\mathbb{C}^d$. This includes, in particular, the case, when $A$ is a $d$-dimensional representation of the group of rotations $SO(3)$. The Hamiltonian is sometimes called the '$GL(d)$-invariant magnet'. In the literature the term 'Heisenberg model' is often reserved for the case $d = 2$.

The operator $P$ also plays in important role for the implementation of the action of spatial symmetries on quantum-spin systems, since it induces the action of the symmetric group $\mathfrak{S}^{2L}$ on $\mathcal{H}_{2L}$. For $j, k, l \in \{-L+1, \ldots, L\}$ mutually distinct

$$P_{jk} e_{k\,\alpha}^{\ \beta} = e_{j\,\alpha}^{\ \beta} P_{jk} , \tag{1.15a}$$

$$P_{jk}^2 = \mathrm{id}\,, \tag{1.15b}$$

$$P_{jk}P_{kl} = P_{jl}P_{jk} = P_{kl}P_{jl}\,, \tag{1.15c}$$

which follows immediately from the definition (1.11) of $P$. The braid relation

$$P_{jj+1}P_{j+1j+2}P_{jj+1} = P_{j+1j+2}P_{jj+1}P_{j+1j+2} \tag{1.16}$$

follows from (1.15b), (1.15c). Braid relation and (1.15b) define the symmetric group.

The symmetry group of a spin chain with an even number of sites is the dihedral group $\mathcal{D}_{2L} = \mathcal{C}_{2L} \rtimes \mathcal{C}_2 \subset \mathfrak{S}^{2L}$ which is the symmetry group of a regular polygon with $2L$ edges. Being a product of two cyclic groups it has two generators

$$\hat{U} = P_{-L+1,-L+2} \ldots P_{L-1,L}\,, \tag{1.17a}$$

$$\hat{P} = P_{-L+1,L}P_{-L+2,L-1} \ldots P_{0,1}\,, \tag{1.17b}$$

the 'shift operator' $\hat{U}$ and the 'parity operator' $\hat{P}$. Note that $\hat{U}^{2L} = \hat{P}^2 = \mathrm{id}$.

Here is a third example for the occurance of $P$ in the theory of quantum-spin systems. It provides a family of rational (or 'Yangian') solutions of the Yang-Baxter equation which is behind the integrability of the $GL(d)$ invariant Hamiltonians (1.13). Define

$$R(\lambda, \mu) = (\lambda - \mu)I_d \otimes I_d + P\,. \tag{1.18}$$

Then, using (1.15c), it is easy to see that

$$R_{jk}(\lambda, \mu)R_{jl}(\lambda, \nu)R_{kl}(\mu, \nu) = R_{kl}(\mu, \nu)R_{jl}(\lambda, \nu)R_{jk}(\lambda, \mu)\,, \tag{1.19}$$

if $j, k, l \in \{-L+1, \ldots, L\}$ are mutually distinct.

## 1.3    Interactions

In the following we shall focus on quantum-spin chains with Hilbert space $\mathcal{H}_{2L}$ and with local interactions $h \in \mathrm{End}(\mathbb{C}^d)^{\otimes m}$, where $m \in \{2, \ldots, 2L\}$ will be called the range of the interaction. Setting

$$h_{j,j+1,\ldots,j+m-1} = \hat{U}^{j-1}h_{1,\ldots,m}\hat{U}^{1-j} \tag{1.20}$$

for $j = -L+1, \ldots, L$ we define the Hamiltonian

$$H = \sum_{j=-L+1}^{L} h_{j,j+1,\ldots,j+m-1} \tag{1.21}$$

which is translation invariant ('satisfies periodic boundary conditions') by construction.

## 1.4    Statistical mechanics of quantum-spin systems

After preparation in an experiment any quantum-spin system will be in a state described by a density matrix (a statistical operator) $\rho_L \in \mathrm{End}\,\mathcal{H}_{2L}$ with properties

$$\rho_L = \rho_L^+\,, \quad \rho_L \le 0\,, \quad \mathrm{tr}\,\rho_L = 1\,. \tag{1.22}$$

These properties guarantee that $\rho_L$ is diagonalizable and that the spectrum of $\rho_L$ is a discrete probability distribution. We may think of $\rho_L$ as representing an ensemble. Subsequent experiments then measure ensemble averages

$$\langle X \rangle = \mathrm{tr}_{-L+1,\ldots,L}\{\rho_L X\} \tag{1.23}$$

of operators $X \in \text{End}_{\mathcal{H}_{2L}}$.

In general $\rho_L$ is time-dependent and its time dependence (in the Schrödinger picture!) is determined by the von-Neumann equation

$$i\partial_t \rho_L = [H, \rho_L]\,. \tag{1.24}$$

Hence, a stationary density matrix should be a function of the conserved quantities commuting with the Hamiltonian.

### 1.4.1 Examples

In these notes we restrict ourselves to stationary density matrices. Some important examples are listed below.

(i) Many-body quantum systems cannot be separated from their environment forever. Eventually they relax to the canonical ensemble,

$$\rho_L(T) = \frac{\mathrm{e}^{-H/T}}{\text{tr}_{-L+1,\dots,L}\{\mathrm{e}^{-H/T}\}}\,. \tag{1.25}$$

However, transients and long relaxation times are possible, particularly for integrable quantum-spin systems, and stationary non-equilibrium ensembles may be realised in driven systems.

(ii) Two more examples are the zero and infinite-temperature limits of $\rho_L(T)$.

$$\lim_{T\to 0+}\rho_L(T) = \frac{1}{g}\sum_{j=1}^{g}|\psi_j^{(0)}\rangle\langle\psi_j^{(0)}|\,, \tag{1.26}$$

where the $\{|\psi_j^{(0)}\rangle\}_{j=1}^{g}$ form an orthonormal basis of the ground-state sector of $\mathcal{H}_{2L}$ and $g$ is the ground-state degeneracy, and

$$\lim_{T\to +\infty}\rho_L(T) = d^{-2L}\cdot\text{id}\,. \tag{1.27}$$

(iii) A special case of ensembles are those represented by any excited state $|\psi_n\rangle$ of the Hamiltonian $H$. The corresponding density matrices are

$$\rho_L^{(n)} = |\psi_n\rangle\langle\psi_n|\,. \tag{1.28}$$

### 1.4.2 Integrable quantum-spin systems

These notes will focus on large integrable quantum-spin systems. The problems we are going to address comprise:

(i) A description of density matrices $\rho_L$ in a way compatible with the integrable structure,

(ii) the calculation of the free energy per lattice site in the thermodynamic limit,

$$f(T) = -T\lim_{L\to +\infty}\frac{1}{L}\ln\big(\text{tr}_{-L+1,\dots,L}\{\mathrm{e}^{-H/T}\}\big)\,, \tag{1.29}$$

(iii) the calculation of two-point functions of local operators in the thermodynamic limit,

$$\langle x_1 y_{m+1}\rangle = \lim_{L\to +\infty}\text{tr}_{-L+1,\dots,L}\big\{\rho_L(T)x_1 y_{m+1}\big\}\,. \tag{1.30}$$

## 1.5 Comments

It is an interesting problem to describe the class of density matrices that may be generated as a result of the relaxation of an isolated integrable quantum spin system towards equilibrium. Due to the existence of a large number of additional local conserved quantities, the class of such density matrices must be much larger than the one-parametric family (1.25). In [7] the concept of generalised Gibbs ensembles was suggested with an inverse-temperature like Lagrange parameter for every additional conserved quantity. Conceptual difficulties arise from the fact that it is hard to identify 'a complete set of conserved local operators' in an infinite integrable system. We recommend [8] for further reading.

# L2  The quantum transfer matrix

## 2.1  Fundamental models

A sufficiently general setting for a statistical mechanics of integrable quantum-spin systems is that of 'fundamental Yang-Baxter integrable models'. Fundamental integrable spin systems are entirely defined in terms of a matrix $R(\lambda, \mu) : \mathbb{C}^2 \mapsto \mathrm{End}(\mathbb{C}^d \otimes \mathbb{C}^d)$ which satisfies

$$R_{12}(\lambda, \mu)R_{13}(\lambda, \nu)R_{23}(\mu, \nu) = R_{23}(\mu, \nu)R_{13}(\lambda, \nu)R_{12}(\lambda, \mu)\,, \tag{2.31a}$$

$$R(\lambda, \lambda) = P\,. \tag{2.31b}$$

Equation (2.31a) is the famous Yang-Baxter equation. It is this equation that underlies the integrability of many quantum-spin systems. Solutions of the Yang-Baxter equation are called $R$-matrices. An $R$-matrix satisfying equation (2.31b) is called regular. The arguments of $R(\lambda, \mu)$ are called spectral parameters.

Assuming differentiability in $\lambda, \mu$ in a vicinity of $(0,0)$ equations (2.31) imply another property of the $R$-matrix which is called unitarity: There is a function $g : \mathbb{C}^2 \mapsto \mathbb{C}$, differentiable in a neighbourhood of $(0,0)$, $g(0,0) = 1$, $g(\lambda, \mu) = g(\mu, \lambda)$, such that

$$\frac{R_{12}(\lambda, \mu)R_{21}(\mu, \lambda)}{g(\lambda, \mu)g(\mu, \lambda)} = \mathrm{id}\,. \tag{2.32}$$

We may therefore assume in the following that $R$ is normalised in such a way that

$$R_{12}(\lambda, \mu)R_{21}(\mu, \lambda) = \mathrm{id}\,. \tag{2.33}$$

The proof of the existence of the function $g$ is left as an exercise to the reader.

With any $R$-matrix satisfying (2.31), (2.33) we associate two transfer matrices

$$t_{\perp}(\lambda) = \mathrm{tr}_a\{R_{a,L}(\lambda, 0)\dots R_{a,-L+1}(\lambda, 0)\}\,, \tag{2.34a}$$

$$\bar{t}_{\perp}(\lambda) = \mathrm{tr}_a\{R_{-L+1,a}(0, \lambda)\dots R_{L,a}(0, \lambda)\} \tag{2.34b}$$

and a Hamiltonian

$$H = h_R t'_{\perp}(0)t_{\perp}^{-1}(0) = -h_R t_{\perp}(0)\bar{t}'_{\perp}(0) = h_R \sum_{j=-L+1}^{L} \partial_\lambda (PR)_{j-1,j}(\lambda, 0)\big|_{\lambda=0}\,, \tag{2.35}$$

where $(PR)_{-L,-L+1} = (PR)_{L,-L+1}$ by definition and where $h_R \in \mathbb{C}$ is a constant which may be used to render $H$ Hermitian and to set the energy scale (alternatively one may rescale the spectral parameter). In order to obtain the second equation in (2.35) one has to use (2.33).

## 2.2    Trotter formula

Let

$$X_N = \frac{t_\perp\left(-\frac{h_R}{2NT}\right)\bar{t}_\perp\left(\frac{h_R}{2NT}\right) - 1}{1/N} \tag{2.36}$$

and observe that, due to (2.35),

$$\lim_{N\to+\infty} X_N = -\frac{H}{T}\,. \tag{2.37}$$

It is not difficult to see that

$$\left\|\mathrm{e}^{-H/T} - \left[t_\perp\left(-\frac{h_R}{2NT}\right)\bar{t}_\perp\left(\frac{h_R}{2NT}\right)\right]^N\right\| \le \left\|\mathrm{e}^{-H/T} - \mathrm{e}^{X_N}\right\| + \frac{\|X_N\|^2}{2N}\,\mathrm{e}^{\|X_N\|}\,, \tag{2.38}$$

where $\|\cdot\|$ is the operator norm. Hence,

$$\mathrm{e}^{-H/T} = \lim_{N\to+\infty}\left[t_\perp\left(-\frac{h_R}{2NT}\right)\bar{t}_\perp\left(\frac{h_R}{2NT}\right)\right]^N\,. \tag{2.39}$$

This way the (unnormalised) density matrix of the canonical ensemble is represented as a product of transfer matrices. Equation (2.39) is sometimes called 'the Trotter formula', $N$ 'the Trotter number'.

## 2.3    External fields

Assume there is $\Theta(\alpha) = \mathrm{e}^{\alpha\hat{\varphi}}$ with $\hat{\varphi} \in \mathrm{End}(\mathbb{C}^d)$, $\alpha \in \mathbb{C}$ such that

$$[R_{12}(\lambda,\mu),\Theta_1(\alpha)\Theta_2(\alpha)] = 0\,, \tag{2.40}$$

which is called a $U(1)$ symmetry of the $R$-matrix. Then

$$[t_\perp(\lambda),\Theta_{-L+1}(\alpha)\dots\Theta_L(\alpha)] = [\bar{t}_\perp(\lambda),\Theta_{-L+1}(\alpha)\dots\Theta_L(\alpha)] = 0\,. \tag{2.41}$$

Setting

$$\hat{\Phi} = \sum_{j=-L+1}^{L}\hat{\varphi}_j \tag{2.42}$$

we conclude that

$$[t_\perp(\lambda),\hat{\Phi}] = [\bar{t}_\perp(\lambda),\hat{\Phi}] = 0\,. \tag{2.43}$$

Setting

$$H_L = H - \kappa\hat{\Phi} \tag{2.44}$$

this allows us to couple an external field to the Hamiltonian without spoiling its integrability.

## 2.4    Quantum transfer matrix

For $N$ even introduce 'vertical spaces' $\bar{1},\dots,\overline{N}$. By definition

$$T_a(\lambda|\alpha) = \Theta_a(\alpha)R^{t_1}_{\overline{N},a}(\nu_N,\lambda)R_{a,\overline{N-1}}(\lambda,\nu_{N-1})\dots R^{t_1}_{\bar{2},a}(\nu_2,\lambda)R_{a,\bar{1}}(\lambda,\nu_1) \tag{2.45}$$

is the 'staggered and twisted inhomogeneous monodromy matrix' of the fundamental model. Here

$$R^{t_1}(\lambda,\mu) = R^{\alpha\gamma}_{\beta\delta}e^\alpha_\beta \otimes e^\delta_\gamma\,. \tag{2.46}$$

The Yang-Baxter equation (2.31a) implies

$$R_{ab}(\lambda, \mu)T_a(\lambda|\alpha)T_b(\mu|\alpha) = T_b(\mu|\alpha)T_a(\lambda|\alpha)R_{ab}(\lambda, \mu) \tag{2.47}$$

(Exercise: Prove it!). These are the 'Yang-Baxter algebra relations' for the (operator-valued) matrix elements of $T_a(\lambda|\alpha)$ considered as a matrix in 'auxiliary space $a$'.

We shall call the transfer matrix associated with $T_a(\lambda|\alpha)$ the 'quantum transfer matrix' of the fundamental model and denote it by

$$t(\lambda|\alpha) = \text{tr}_a\{T_a(\lambda|\alpha)\}. \tag{2.48}$$

Using (2.39) and the Yang-Baxter equation (2.31a) we obtain

$$\mathrm{e}^{-H_L/T} = \mathrm{e}^{\kappa\hat{\Phi}/T}\,\mathrm{e}^{-H/T} = \lim_{N\to\infty} \mathrm{e}^{\kappa\hat{\Phi}/T}\Big[t_\perp\Big(-\frac{h_R}{NT}\Big)\bar{t}_\perp\Big(\frac{h_R}{NT}\Big)\Big]^{\frac{N}{2}}$$

$$= \lim_{N\to\infty} \text{tr}_{\bar{1},\dots,\overline{N}}\{T_{-L+1}(0)\dots T_L(0)\} \quad (2.49)$$

if we set $\nu_{2j-1} = h_R/(NT)$, $\nu_{2j} = -h_R/(NT)$, $j = 1, \dots, N/2$, and $\alpha = \kappa/T$.

It is instructive (and not too hard) to derive (2.49) by algebraic means (see e.g. [6]). In the following, however, we shall introduce the graphical language of vertex models and use it for an intuitive and easily memorizable proof.

## 2.5 Example: the XXZ chain

The basic example of a fundamental integrable model is the XXZ spin-$\frac{1}{2}$ chain. Its $R$-matrix (the $R$-matrix of the six-vertex model [9]) can be understood as a '$q$-deformation' of the rational $R$-matrix (1.18) for $d = 2$. With a rescaling appropriate for our purposes it becomes

$$R(\lambda, \mu) = \begin{pmatrix} 1 & 0 & 0 & 0 \\ 0 & b(\lambda - \mu) & c(\lambda - \mu) & 0 \\ 0 & c(\lambda - \mu) & b(\lambda - \mu) & 0 \\ 0 & 0 & 0 & 1 \end{pmatrix}, \tag{2.50a}$$

$$b(\lambda) = \frac{\text{sh}(\lambda)}{\text{sh}(\lambda + \eta)}, \quad c(\lambda) = \frac{\text{sh}(\eta)}{\text{sh}(\lambda + \eta)}. \tag{2.50b}$$

It is a simple exercise to verify that (2.50) describes a one-parameter family of solutions of the Yang-Baxter equation (2.31a) that is regular (2.31b) and unitary (2.33).

In order to generate the Hamiltonian we differentiate

$$\partial_\lambda PR(\lambda, 0)\big|_{\lambda=0} = \begin{pmatrix} 0 & & & \\ & c'(0) & b'(0) & \\ & b'(0) & c'(0) & \\ & & & 0 \end{pmatrix} = \frac{1}{\text{sh}(\eta)}\begin{pmatrix} 0 & & & \\ & -\Delta & 1 & \\ & 1 & -\Delta & \\ & & & 0 \end{pmatrix}$$

$$= \frac{1}{2\,\text{sh}(\eta)}\{\sigma^x \otimes \sigma^x + \sigma^y \otimes \sigma^y + \Delta(\sigma^z \otimes \sigma^z - \text{id})\}, \quad (2.51)$$

where $\sigma^\alpha$, $\alpha = x, y, z$, are Pauli matrices and $\Delta = \text{ch}(\eta)$ by definition. Setting $h_R = 2J\,\text{sh}(\eta)$ in our general formula (2.35) we obtain the XXZ Hamiltonian

$$H_{XXZ} = J\sum_{j=-L+1}^{L}\{\sigma_{j-1}^x\sigma_j^x + \sigma_{j-1}^y\sigma_j^y + \Delta(\sigma_{j-1}^z\sigma_j^z - 1)\}, \tag{2.52}$$

where $\sigma^\alpha_{-L} = \sigma^\alpha_L$ by definition. $H_{XXZ}$ is hermitian for all real $J$ and $\Delta$. A closer inspection of its discrete symmetries reveals that we may restrict ourselves to $J > 0$.

Clearly

$$[R(\lambda, \mu), \mathrm{e}^{\alpha \sigma^z/2} \otimes \mathrm{e}^{\alpha \sigma^z/2}] = 0 \,, \tag{2.53}$$

meaning that $R(\lambda, \mu)$ has a $U(1)$-symmetry generated by $\Theta(\alpha) = \mathrm{e}^{\alpha \hat\varphi}$ with $\hat\varphi = \sigma^z/2$. Thus, in this case

$$H_L = H_{XXZ} - \frac{\kappa}{2} \sum_{j=-L+1}^{L} \sigma^z_j \tag{2.54}$$

(cf. (2.44)) and $\kappa$ has the meaning of a Zeeman magnetic field coupling to the individual spins.

## 2.6 Graphical representation of integrability objects

A rather efficient way of dealing with the relation between the various 'integrability objects' introduced above, the $R$-matrix and the transfer and monodromy matrices, utilises a certain graphical representation [9].

(i) We identify every matrix element of $R(\lambda, \mu)$ with a 'vertex',

$$R^{\alpha\gamma}_{\beta\delta}(\lambda, \mu) \;=\; \beta \underset{\lambda}{\underrightarrow{\qquad}} \alpha \,, \tag{2.55}$$

$\alpha, \beta, \gamma, \delta = 1, \ldots, d$ (for $d = 2$ also $\pm$ or $\uparrow, \downarrow$).

(ii) Every arrangement of a finite number of directed, crossing lines is then in one-to-one correspondence with a product of $R$-matrix elements. We identify the connection of lines with summation over indices, e.g.

$$\beta \underset{\lambda}{\underrightarrow{\qquad}} \alpha \;=\; R^{\alpha\gamma_2}_{\beta'\delta_2}(\lambda, \nu) R^{\beta'\gamma_1}_{\beta\delta_1}(\lambda, \mu) \,. \tag{2.56}$$

(iii) We indicate closed lines by a small semi-loop at the tail, e.g.

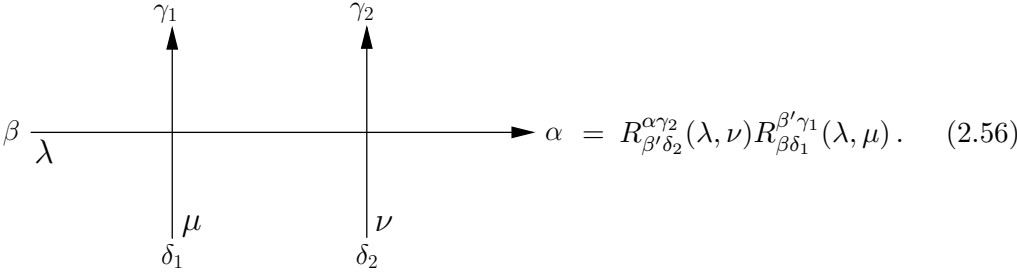

$$\beta \underset{\lambda}{\underrightarrow{\qquad}} \alpha \;=\; R^{\alpha\gamma}_{\beta\gamma}(\lambda, \mu) \;=\; \beta \underset{\lambda}{\underrightarrow{\qquad}} \alpha \,. \tag{2.57}$$

(iv) This way we obtain a graphical representation of the Yang-Baxter equation:

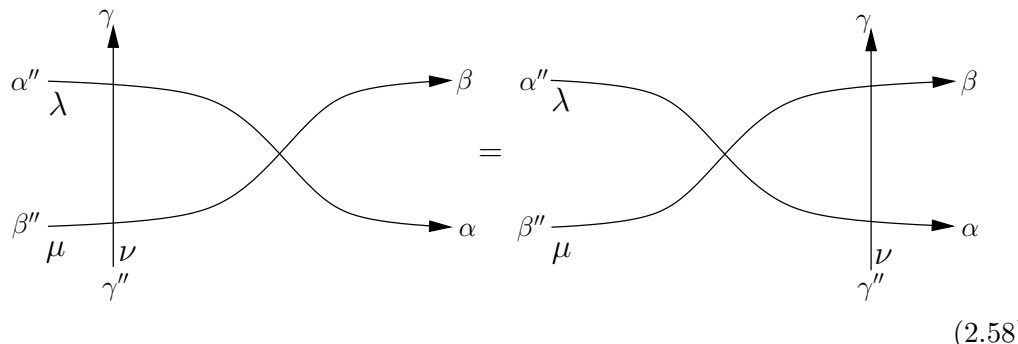

$$(2.58)$$

translates into

$$R^{\alpha\beta}_{\alpha'\beta'}(\lambda,\mu)R^{\alpha'\gamma}_{\alpha''\gamma'}(\lambda,\nu)R^{\beta'\gamma'}_{\beta''\gamma''}(\mu,\nu) = R^{\beta\gamma}_{\beta'\gamma'}(\mu,\nu)R^{\alpha\gamma'}_{\alpha'\gamma''}(\lambda,\nu)R^{\alpha'\beta'}_{\alpha''\beta''}(\lambda,\mu) \qquad (2.59)$$

which is the coordinate form of the Yang-Baxter equation (2.31a) (Exercise: Verify!).

(v) For consistency we need the rule

$$\beta \xrightarrow{\quad\lambda\quad} \alpha \;=\; \delta^\alpha_\beta\,. \qquad (2.60)$$

(vi) Then regularity (2.31b) has the graphical representation

$$(2.61)$$

(vii) And unitarity is drawn as

$$(2.62)$$

(Exercise: Show (vi) and (vii)!)

(viii) Single-site operators, such as $\Theta(\kappa)$ can be represented as

$$\Theta^\alpha_\beta(\kappa) \;=\; \beta \xrightarrow[\lambda]{\;\;\kappa\;\;\times\;\;} \alpha\,. \qquad (2.63)$$

Then the $U(1)$ symmetry (2.40) becomes

$$\beta \xrightarrow{\lambda} \alpha \;=\; \Theta^{\alpha}_{\alpha'}(\kappa)\Theta^{\gamma}_{\gamma'}(\kappa)R^{\alpha'\gamma'}_{\beta\delta}(\lambda,\mu)$$

$$=\; R^{\alpha\gamma}_{\beta'\delta'}(\lambda,\mu)\Theta^{\beta'}_{\beta}(\kappa)\Theta^{\delta'}_{\delta}(\kappa) \;=\; \beta \xrightarrow{\lambda} \alpha \;. \quad (2.64)$$

(ix) Examples:

$$R^{t_1\,\gamma\alpha}_{\quad\delta\beta}(\nu,\lambda) \;=\; R^{\delta\alpha}_{\gamma\beta}(\nu,\lambda) \;=\; \beta \xrightarrow{\lambda} \alpha \qquad\qquad (2.65)$$

$$\Theta^{\alpha}_{\alpha'}(\kappa)R^{t_1\,\gamma_2\alpha'}_{\quad\delta_2\alpha''}(\nu_2,\lambda)R^{\alpha''\gamma_1}_{\beta\delta_1}(\lambda,\nu_1) \;=\; \beta \xrightarrow{\lambda} \alpha$$

$$(2.66)$$

$$\left(\bar{t}_{\perp}(\lambda)\right)^{\gamma_{-L+1}\ldots\gamma_L}_{\delta_{-L+1}\ldots\delta_L} = R^{\gamma_{-L+1}\alpha_{-L+1}}_{\delta_{-L+1}\alpha_{-L+2}}(0,\lambda)R^{\gamma_{-L+2}\alpha_{-L+2}}_{\delta_{-L+2}\alpha_{-L+3}}(0,\lambda)\ldots R^{\gamma_L\alpha_L}_{\delta_L\alpha_{-L+1}}(0,\lambda)$$

$$=\qquad\qquad\qquad\qquad\qquad\qquad\qquad (2.67)$$

$$\left(t_{\perp}(\lambda)\bar{t}_{\perp}(\mu)\right)^{\gamma_{-L+1}\ldots\gamma_L}_{\delta_{-L+1}\ldots\delta_L} \;=\;\qquad\qquad\qquad\qquad\qquad . \quad (2.68)$$

## 2.7  Comments

The graphical notation is generally useful in tensor calculus, not only in integrable systems. Other examples are Feynman diagrams, tensor networks and matrix product states.

# L3    Partition function, density matrix and static correlation functions

## 3.1    Statistical operator of the (grand) canonical ensemble

The graphical notation is appropriate for discussing the various types of (reduced) density matrices considered in the literature. We start with the most fundamental one which is the statistical operator of the grand canonical ensemble. Define

$$\widetilde{\rho}_{N,L} = \mathrm{e}^{\kappa \hat{\Phi}/T} \Big[ t_\perp \Big( -\frac{h_R}{NT} \Big) \bar{t}_\perp \Big( \frac{h_R}{NT} \Big) \Big]^{\frac{N}{2}} , \quad Z_{N,L} = \mathrm{tr}_{-L+1,\dots,L}\, \widetilde{\rho}_{N,L} . \tag{3.69}$$

Since, by (2.49),

$$\lim_{N\to\infty} \widetilde{\rho}_{N,L} = \mathrm{e}^{-H_L/T} , \tag{3.70}$$

we call

$$\rho_{N,L} = \frac{\widetilde{\rho}_{N,L}}{Z_{N,L}} \tag{3.71}$$

the finite Trotter number approximant to the statistical operator.

$\widetilde{\rho}_{N,L}$ has the graphical representation

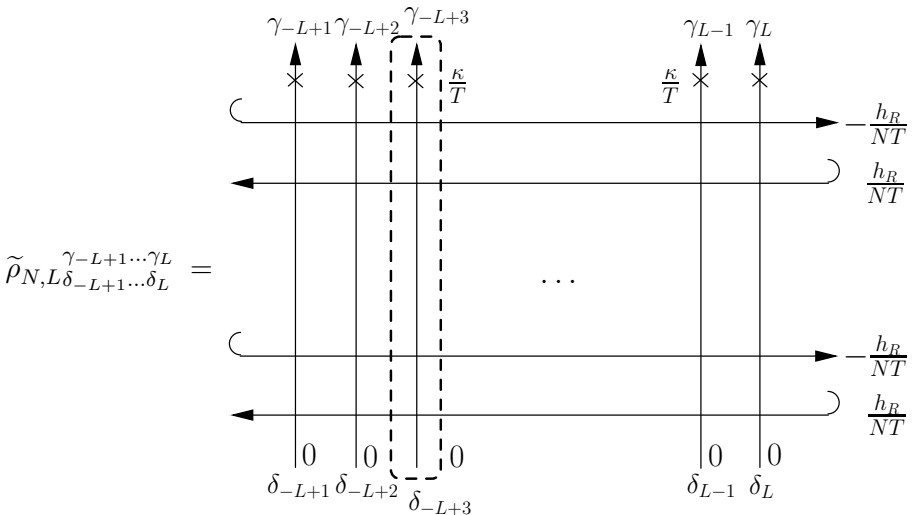

$$= \mathrm{tr}_{\bar{1},\dots,\overline{N}}\big\{ T_{-L+1}(0) \dots T_L(0) \big\} . \tag{3.72}$$

Regarding the graph directly and in a reference frame rotated by $\pi/2$, the equality of left and right hand side (an hence the proof of (2.49)) becomes obvious and needs no further explanation. Note that the object in the dashed frame is the staggered, inhomogeneous monodromy matrix introduced above.

## 3.2    Partition function and free energy per lattice site

The thermodynamics of a quantum spin system is determined by its partition function

$$Z_L = \mathrm{tr}_{-L+1,\dots,L}\big\{ \mathrm{e}^{-H_L/T} \big\} = \lim_{N\to\infty} Z_{N,L} = \lim_{N\to\infty} \mathrm{tr}_{\bar{1},\dots,\overline{N}}\big\{ \big( t(0|\kappa/T) \big)^{2L} \big\} \tag{3.73}$$

or by its free energy

$$F_L = -T \ln Z_L , \tag{3.74}$$

respectively.

Let us denote the eigenvalues of $t(\lambda|\kappa/T)$ by $\Lambda_n(\lambda|\kappa)$, $n = 0, \ldots, d^N - 1$, and assume that they are ordered in such a way that

$$|\Lambda_0(0|\kappa)| \geq |\Lambda_1(0|\kappa)| \geq |\Lambda_2(0|\kappa)| \geq \ldots . \tag{3.75}$$

If

(i) the limits $N \to \infty$ and $L \to \infty$ commute and

(ii) $|\Lambda_0(0|\kappa)/\Lambda_n(0|\kappa)| < 1$ for all $N \in 2\mathbb{Z}_+$ and all $n = 1, \ldots, d^N - 1$,

the expression of the free energy per lattice site greatly simplifies in the thermodynamic limit:

$$f(T, \kappa) = \lim_{L \to \infty} \frac{F_L}{2L} = -T \lim_{L \to \infty} \lim_{N \to \infty} \frac{1}{2L} \ln\left\{ \sum_{n=0}^{d^N - 1} \Lambda_n^{2L}(0|\kappa) \right\}$$

$$= -T \lim_{N \to \infty} \ln \Lambda_0(0|\kappa) . \tag{3.76}$$

It is determined by a sequence of non-degenerate eigenvalues $\Lambda_0$ of the quantum transfer matrix. We shall call these eigenvalues of largest modulus for fixed $N$ the dominant eigenvalues, the corresponding eigenvectors $|\kappa\rangle$ the dominant eigenvectors.

Proving statements such as the commutativity of the Trotter limit and the thermodynamic limit usually falls into the realm of mathematical analysis. Physicists often take a pragmatic attitude toward more sophisticated mathematical questions, assuming everything to be alright until a counterexample appears. This is perhaps why statements (i) and (ii) above have not yet been rigorously justified. Still we have good reasons to believe that they are true for all fundamental integrable models and for all $T > 0$.

(i) Equation (3.76) was used to study the thermodynamics of many fundamental integrable lattice models, most notably of the Heisenberg XXX and XXZ spin chains [5,10] and of the Hubbard model [11]. These studies have been compared with other independent methods and give the same results within the available numerical accuracy. They reproduce the correct high and low-temperature behaviour and have the correct free Fermion limits in those cases where they exist.

(ii) For the special case of the XXZ chain a rigorous proof was recently provided for high enough finite temperatures [12]. Since the proof uses basically only the locality of the interaction of the model, it is expected to be generalizable at least to all fundamental integrable models.

(iii) The commutativity of limits was proved for a slightly differently defined quantum transfer matrix (which is unfortunately less compatible with the integrable structure imposed by the Yang-Baxter equation) by M. Suzuki in 1985 [4].

We would like to point out that the existence of the first limit in (3.76), defining the free energy per lattice site, was proved long time ago (see e.g. [13]). Thus, the commutativity of the limits would also imply the existence of the limit on the right hand side of the equation.

**Remark.** So far we did not use the integrability in any essential way. What we used was

(i) the $U(1)$ symmetry

$$[H, \Phi] = 0 \tag{3.77}$$

and the regularity

$$R(\lambda, \mu) = P + (\lambda - \mu)PH^{(2)} + \ldots \tag{3.78}$$

where $H^{(2)}$ is a two-site Hamiltonian and the dots denote terms quadratic in $\lambda$ and $\mu$.

Simply defining for a given two-site Hamiltonian $H^{(2)}$

$$R_{H^{(2)}}(\lambda, \mu) = P + (\lambda - \mu)PH^{(2)} \tag{3.79}$$

and using the $U(1)$ symmetry of this object if it exists we obtain a quantum transfer matrix with dominant eigenvalue $\Lambda_0(0|\kappa)$ and (3.76) remains valid. Equation (3.76) can then be used as the starting point for the implementation of a numerical algorithm for the calculation of thermodynamic properties of infinite spin chain systems [14]. The integrability enters the game only when we calculate $\Lambda_0$.

## 3.3 Density matrix of a chain segment (reduced density matrix) and expectation values of local operators

For any $A \in \operatorname{End} \mathcal{H}_{2L}$ other than the identity operator there exist $k, l \in \{-L+1, \dots, L\}$, $k \leq l$ and $X \in \operatorname{End}\big(\mathbb{C}^d\big)^{\otimes(l-k+1)}$ such that $A$ acts non-trivially on sites $k$ and $l$ and

$$A = X_{k,k+1,\dots,l} . \tag{3.80}$$

$X$ is the non-trivial part of $A$. We identify the chain segment associated with lattice sites $k, \dots, l$ (a certain number of factors in $\big(\mathbb{C}^d\big)^{\otimes 2L}$) with the 'interval' $[k, l] = (k, k+1, \dots, l)$ and write

$$X_{k,k+1,\dots,l} = X_{[k,l]} . \tag{3.81}$$

The number of sites in $[k, l]$ will be called the length of $X$, $\ell(X)$. Here $\ell(X) = l - k + 1$. These notions still make sense for $L \to \infty$. An operator, whose length stays finite for $L \to \infty$ will be called local. For any local operator of length $m$, with non-trivial part $X$ on the infinite chain its grand canonical expectation value is

$$\langle X_{[k,l]} \rangle = \lim_{L\to\infty} \operatorname{tr}_{-L+1,\dots,L}\big\{\rho_L(T)X_{[k,l]}\big\} = \lim_{L\to\infty} \operatorname{tr}_{-L+1,\dots,L}\big\{\rho_L(T)X_{[1,\ell(X)]}\big\}$$

$$= \lim_{L\to\infty} \lim_{N\to\infty} \operatorname{tr}_{-L+1,\dots,L}\big\{\rho_{N,L}X_{[1,m]}\big\}$$

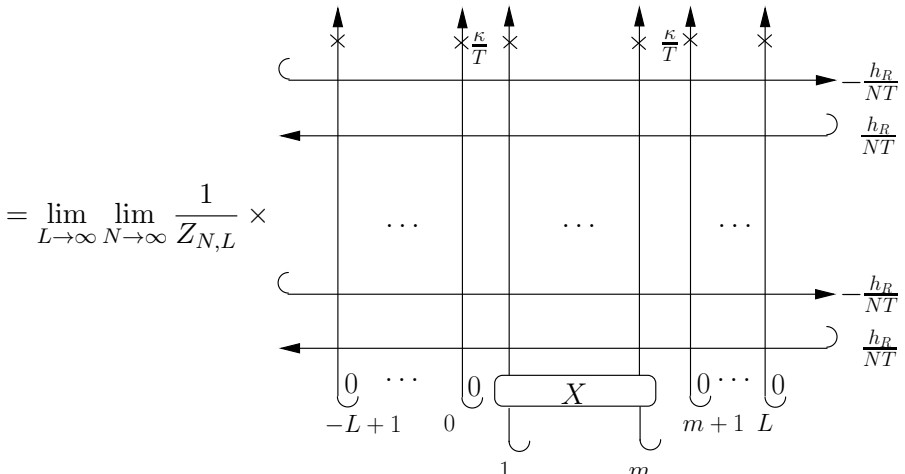

$$= \lim_{L\to\infty} \lim_{N\to\infty} \frac{1}{Z_{N,L}} \times$$

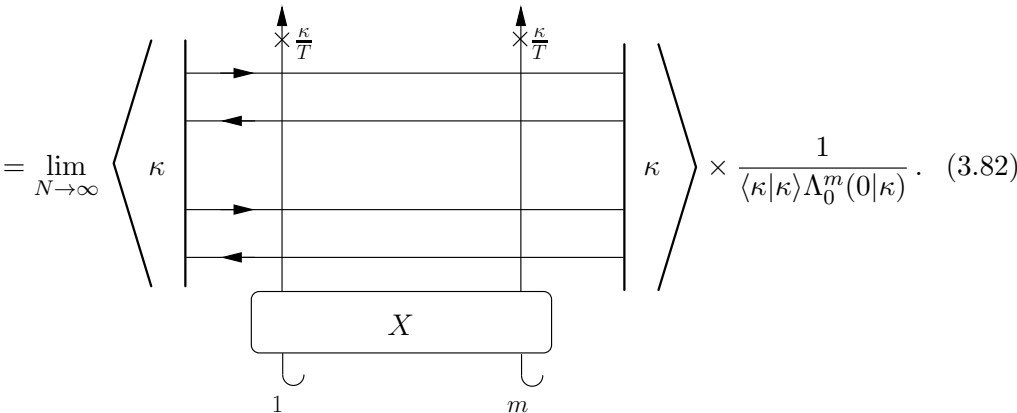

$$= \lim_{N \to \infty} \left\langle \kappa \left| \quad X \quad \right| \kappa \right\rangle \times \frac{1}{\langle \kappa | \kappa \rangle \Lambda_0^m(0|\kappa)} . \quad (3.82)$$

Here we have used the translation invariance of the Hamiltonian in the second equation, have inserted the finite Trotter number approximant to the statistical operator in the third equation, have represented the resulting expression graphically in the fourth equation, and have used the fact that the quantum transfer matrix projects on its dominant state, if applied many times, in the fifth equation. The latter property holds under the assumption that points (i) and (ii) below equation (3.75) are satisfied. The box around the letter '$X$' represents the corresponding operator. For the graphical representation of the dominant state and its dual we use the self-explanatory symbols $|\kappa\rangle$ and $\langle\kappa|$. We see that the expectation values of all local operators of length $m$ (on the infinite chain) can be calculated by means of the finite Trotter number approximant to the reduced density matrix associated with the interval (or chain chain segment) $[1, m]$,

$$D_m^{(N)}(T, \kappa) = \left\langle \kappa \left| \quad \cdots \quad \right| \kappa \right\rangle \times \frac{1}{\langle \kappa | \kappa \rangle \Lambda_0^m(0|\kappa)} . \quad (3.83)$$

The reduced density matrix $D_m(T, \kappa)$ is obtained in the Trotter limit $N \to \infty$, and, as can be read off from (3.82),

$$\langle X_{[k,l]} \rangle = \text{tr}_{1,\dots,m} \{ D_m(T, \kappa) X \} . \quad (3.84)$$

Using the sequence $D_m(T, \kappa)_{m \in \mathbb{N}}$ of reduced density matrices we can calculate the expectation value of any local operator on the infinite chain (a procedure which is called an 'inductive limit'). Equation (3.83) is central for the theory of correlation functions of integrable models, in particular of the XXZ chain to be considered in more detail below.

## 3.4   Comments

Coming back to algebraic expressions we see that

$$D_{m+1}(T, \kappa) = \lim_{N \to \infty} \frac{\langle \kappa | T(0|\kappa/T)^{\otimes(m+1)} | \kappa \rangle}{\langle \kappa | \kappa \rangle \Lambda_0(0|\kappa)^{(m+1)}} . \quad (3.85)$$

For any two ultra-local operators $x, y \in \operatorname{End} \mathbb{C}^d$ we define $X(\lambda|\kappa) = \operatorname{tr}\{xT(0|\kappa/T)\}$ and $Y(\lambda|\kappa) = \operatorname{tr}\{yT(0|\kappa/T)\}$. Employing this notation and using (3.84) and (3.85) we obtain the following expression for the two-point correlation function of $x$ and $y$:

$$
\begin{aligned}
\langle x_1 y_{m+1}\rangle &= \lim_{N\to+\infty} \frac{\langle\kappa|\operatorname{tr}\{X(0|\kappa)\}t(0|\kappa/T)^{m-1}\operatorname{tr}\{Y(0|\kappa)\}|\kappa\rangle}{\langle\kappa|\kappa\rangle\Lambda_0(0|\kappa)^m} \\
&= \lim_{N\to+\infty}\sum_n \frac{\langle\kappa|X(0|\kappa)|\kappa,n\rangle\langle\kappa,n|Y(0|\kappa)|\kappa\rangle}{\langle\kappa|\kappa\rangle\Lambda_0(0|\kappa)\langle\kappa,n|\kappa,n\rangle\Lambda_n(0|\kappa)}\left(\frac{\Lambda_n(0|\kappa)}{\Lambda_0(0|\kappa)}\right)^m.
\end{aligned}
\tag{3.86}
$$

In the second line we have expanded $t(0|\kappa/T)^{m-1}\operatorname{tr}\{Y(0|\kappa)\}|\kappa\rangle$ in an eigenbasis $\{|\kappa,n\rangle\}$ of $t(0|\kappa/T)$. The series on the right hand side is what we call a 'thermal form factor series' [15]. It is a useful tool for analysing the large-distance asymptotic behaviour of the two-point function.

## L4  The characterisation of reduced density matrices

### 4.1  Expectation values of local operators in pure states

Using the regularity relation (2.31b) we obtain the following graphical identities

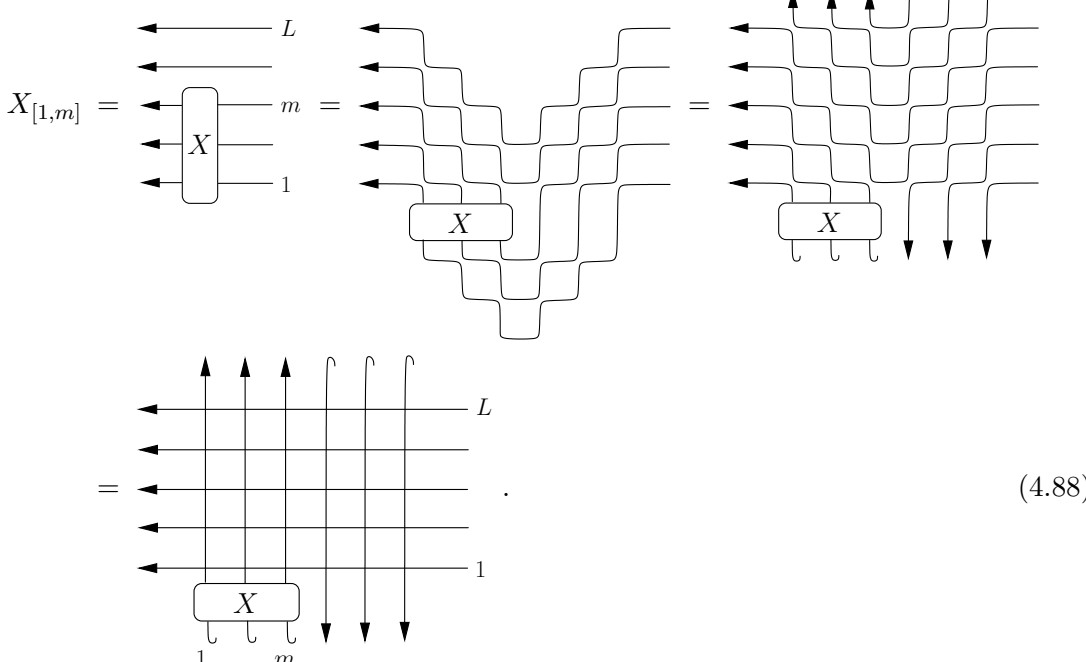

$$\tag{4.87}$$

This generalises to

$$\tag{4.88}$$

If now $|\Psi\rangle$ is any eigenstate of $t_\perp(\lambda)$ with eigenvalue $\Lambda(\lambda)$, then (4.88) implies that

$$\frac{\langle\Psi|X_{[1,m]}|\Psi\rangle}{\langle\Psi|\Psi\rangle} = \left\langle\Psi\left|\begin{array}{c} \\ X \\ \end{array}\right|\Psi\right\rangle \times \frac{1}{\langle\Psi|\Psi\rangle} = \left\langle\Psi\left|\begin{array}{c} \\ X \\ \end{array}\right|\Psi\right\rangle \times \frac{1}{\langle\Psi|\Psi\rangle\Lambda^m(0)} \ . \quad (4.89)$$

These are two graphical representations of the reduced density matrix of the interval $[1,m]$ associated with the eigenstate $|\Psi\rangle$. Remarkably, the expression on the right hand side is of the same form as in (3.82) (a twist can also be included).

**Remark.** We would like to emphasise the following:

  (i) We still did not use integrability here. The construction works for non-integrable models as well.

 (ii) The above is a graphical version of 'the solution of the inverse problem' [16,17].

(iii) The problem of calculating the reduced density matrix from (3.83) or (4.89) is still largely unsolved, even for integrable models. Most of the results available in the literature refer to a few simple example systems related to the spin-$\frac{1}{2}$ XXZ chain on which we concentrate in the following.

## 4.2 Further generalisations of the reduced density matrix with the example of the XXZ chain

An idea in the spirit of Baxter [9], which was rather helpful for actually calculating the reduced density matrix of a chain segment, was to generalise the definition by attaching spectral parameters to the vertical lines [18,19] and a twist to one of the states [20]. We shall denote the corresponding un-normalised reduced density matrix by

$$\mathcal{D}^{(N)}(\xi_1,\ldots,\xi_m|T,\kappa,\kappa') = \left\langle\kappa'\left|\begin{array}{c} \vdots \quad \cdots \quad \vdots \\ \end{array}\right|\kappa\right\rangle \quad (4.90)$$

With this we may define the twisted, inhomogeneous finite Trotter number approximant to the reduced density matrix,

$$D^{(N)}(\xi_1,\ldots,\xi_m|T,\kappa,\kappa') = \frac{\mathcal{D}^{(N)}(\xi_1,\ldots,\xi_m|T,\kappa,\kappa')}{\mathrm{tr}_{1,\ldots,m}\{\mathcal{D}^{(N)}(\xi_1,\ldots,\xi_m|T,\kappa,\kappa')\}} \ . \quad (4.91)$$

It determines the physical reduced density matrix as

$$D_m(T,\kappa) = D(0,\ldots,0|T,\kappa,\kappa) \ , \quad (4.92a)$$

$$D(\xi_1, \ldots, \xi_m | T, \kappa, \kappa') = \lim_{N \to \infty} D^{(N)}(\xi_1, \ldots, \xi_m | T, \kappa, \kappa'). \qquad (4.92b)$$

It turns out that the new parameters $\xi_j$, $j = 1, \ldots, m$, and $\alpha = (\kappa' - \kappa)/T$ regularise the mathematical expressions for $D^{(N)}$. The parameter $\alpha$ acquires a physical meaning in certain scaling limits, e.g. in the conformal limit [21].

For the XXZ spin-$\frac{1}{2}$ chain $D^{(N)}(\xi_1, \ldots, \xi_m | T, \kappa, \kappa')$ has been calculated and characterised in several different ways (for finite $N$ and in the Trotter limit).

(i) By an $m$-fold integral [22].

(ii) By 'discrete rqKZ equations' [23].

(iii) By an 'exponential form', involving a double integral and the annihilation part of the so-called Fermionic basis [24].*

(iv) By the 'JMS theorem' using the creation part of the Fermionic basis [25].

Every single of these charactersations is technically involved and would justify a series of lectures on its own. For this reason we can only provide a brief description, which will necessarily stay somewhat vague, and a short guide to the literature.

The first explicit description of the reduced density matrix of a chain segment of length $m$ of the XXZ chain was obtained in [18] and had form of an $m$-fold multiple integral. It was derived for the ground state of the spin chain in the massive antiferromagnetic regime. The derivation relied on the construction of representations of a deformed vertex-operator algebra.

Subsequently the multiple-integral formula was rederived and generalised by different methods and different authors. An extension to the massless groundstate phase at vanishing magnetic field was obtained in [19]. The derivation relied on the use of functional equations of qKZ-type [26, 27] that had been introduced before in order to characterise form factors of integrable massive quantum field theories.

A derivation by Bethe Ansatz [28] made it possible to take into account a finite magnetic field and opened the way to treat the finite temperature and finite length cases in [29, 30]. A multiple-integral representation for the most general inhomogeneous and twisted case (4.92b) was eventually obtained in [22]. The non-vanishing density matrix elements are of the form

$$D^{\varepsilon'_1 \ldots \varepsilon'_m}_{\varepsilon_1 \ldots \varepsilon_m}(\xi_1, \ldots, \xi_m | T, \kappa, \kappa') = \left[ \prod_{j=1}^{p} \int_{\mathcal{C}} dm(\lambda_j) \, F^{+}_{\ell_j}(\lambda_j) \right] \left[ \prod_{j=p+1}^{m} \int_{\mathcal{C}} d\overline{m}(\lambda_j) \, F^{-}_{\ell_j}(\lambda_j) \right]$$

$$\frac{\det_{j,k=1,\ldots,m} \left[ -G(\lambda_j, \xi_k | \kappa, \kappa') \right]}{\prod_{1 \le j < k \le m} \operatorname{sh}(\lambda_j - \lambda_k - \eta) \operatorname{sh}(\xi_k - \xi_j)}, \qquad (4.93)$$

where we have used the notation

$$dm(\lambda) = \frac{d\lambda}{2\pi i \, \rho(\lambda | \kappa, \kappa')(1 + \mathfrak{a}(\lambda, \kappa))}, \quad d\overline{m}(\lambda) = \mathfrak{a}(\lambda, \kappa) dm(\lambda), \qquad (4.94)$$

$$F^{\pm}_{\ell_j}(\lambda) = \prod_{k=1}^{\ell_j - 1} \operatorname{sh}(\lambda - \xi_k) \prod_{k=\ell_j+1}^{m} \operatorname{sh}(\lambda - \xi_k \mp \eta), \quad \ell_j = \begin{cases} \varepsilon^+_j & j = 1, \ldots, p \\ \varepsilon^-_{m-j+1} & j = p+1, \ldots, m \end{cases}$$

---

*This so far only works in the restricted situation, when the magnetic field $h = 0$. The general case would require the completion of the algebra by a bosonic annihilation operator postulated in [22].

with $\varepsilon_j^+$ the $j$th plus in the sequence $(\varepsilon_j)_{j=1}^m$, $\varepsilon_j^-$ the $j$th minus sign in the sequence $(\varepsilon_j')_{j=1}^m$ and $p$ the number of plus signs in $(\varepsilon_j)_{j=1}^m$.

The definition of the integration contour $\mathcal{C}$ depends on which parameter regime is considered. For $|\Delta| < 1$, for instance, we may choose a rectangle centered around the origin of the complex plane with sides parallel to the real and imaginary axes and of height $|\eta| - 0_+$ and large finite width. The functions $\mathfrak{a}$ and $G$ are solutions of convolution type integral equations with respect to the contour $\mathcal{C}$ and integration kernel

$$K_\alpha(\lambda) = \mathrm{e}^{-\alpha}\,\mathrm{cth}(\lambda - \eta) - \mathrm{e}^\alpha\,\mathrm{cth}(\lambda + \eta)\,. \tag{4.95}$$

The function $\mathfrak{a}$ solves the non-linear integral equation

$$\ln(\mathfrak{a}(\lambda,\kappa)) = -\frac{2\kappa}{T} - \frac{2J\,\mathrm{sh}^2(\eta)}{T\,\mathrm{sh}(\lambda)\,\mathrm{sh}(\lambda+\eta)} - \int_\mathcal{C}\frac{\mathrm{d}\mu}{2\pi\mathrm{i}}K_0(\lambda-\mu)\ln(1+\mathfrak{a}(\mu,\kappa))\,, \tag{4.96}$$

whereas $G$ is the solution of the linear integral equation

$$G(\lambda,\xi|\kappa,\kappa') = \mathrm{e}^{-\alpha}\,\mathrm{cth}(\lambda-\xi-\eta) - \rho(\xi|\kappa,\kappa')\,\mathrm{cth}(\lambda-\xi)$$
$$+ \int_\mathcal{C}\mathrm{d}m(\mu)K_\alpha(\lambda-\mu)G(\mu,\xi)|\kappa,\kappa')\,. \tag{4.97}$$

Here and in (4.94)

$$\rho(\xi|\kappa,\kappa') = \frac{\Lambda_0(\xi|\kappa')}{\Lambda_0(\xi|\kappa)} \tag{4.98}$$

is the ratio of dominant eigenvalues with different twist parameters.

The multiple-integral formula (4.93) is compact and memorizable but, as any true multiple integral, not very efficient for the actual computation of the density matrix elements [31]. Remarkably, however, it turned out that the multiple integrals factorise into sums over products of single integrals. This was explictly worked out with examples [22, 32–34]. It motivated efforts to directly calculate the density matrix elements in factorised form. This is what is the main point behind items (ii)-(iv) above. In [35, 36] a reduced form of the qKZ equation (rqKZ) was derived for the ground state of the XXX and XXZ chains and solved in a form corresponding to the factorised integrals. In [23] the equation was generalised to the finite temperature case. The effort to uncover the structure behind the factorisation led to the discovery of the Fermionic basis in [24, 37] and eventually to a proof of the factorisation under very general conditions ('JMS theorem' [25]). The latter claims that all density matrix elements and hence all static correlation functions can be expressed in terms of only two basic functions, the function $\rho$, equation (4.98), and a function $\omega$ which for the inhomogeneous finite temperature case was characterised in terms of the functions $\mathfrak{a}$ and $G$, equations (4.96), (4.97), in [22]:

$$\mathrm{e}^{-\alpha(\xi_1-\xi_2)}\,\omega(\xi_1,\xi_2|\kappa,\kappa') = 2\Psi(\xi_1,\xi_2|\kappa,\kappa') + K_\alpha(\xi_1-\xi_2)$$
$$+ \big(\rho(\xi_1|\kappa,\kappa') - \rho(\xi_2|\kappa,\kappa')\big)\,\mathrm{cth}(\xi_1-\xi_2)\,, \tag{4.99}$$

where

$$\Psi(\xi_1,\xi_2) = \int_\mathcal{C}\mathrm{d}m(\lambda)\,G(\lambda,\xi_2|\kappa,\kappa')$$
$$\times \big(\mathrm{e}^\alpha\,\mathrm{cth}(\lambda-\xi_1-\eta) - \rho(\xi_1|\kappa,\kappa')\,\mathrm{cth}(\lambda-\xi_1)\big)\,. \tag{4.100}$$

Factorisation has been used to calculate short-range static correlation functions of the XXZ and XXX chains (see e.g. [38, 39] and references listed therein). Here is an example for the XXX chain at $T = 0$, $h = 0$ that was first obtained in [33],

$$D_{41111}^{1111}(0,0) =$$
$$\frac{1}{5} - 2\ln 2 + \frac{173}{60}\zeta(3) - \frac{11}{6}\zeta(3)\ln 2 - \frac{51}{80}\zeta^2(3) - \frac{55}{24}\zeta(5) + \frac{85}{24}\zeta(5)\ln 2\,. \quad (4.101)$$

The Riemann $\zeta$-functions arise from the function $\omega$ in the necessary limits. A more generic example of a finite temperature correlation function of the XXZ chain is

$$\langle \sigma_1^x \sigma_2^x \rangle = -\frac{\omega(0,0|\kappa,\kappa)}{2\,\mathrm{sh}(\eta)} + \frac{\mathrm{ch}(\eta)\partial_\lambda\partial_\alpha\omega(\lambda,0|\kappa+\alpha T,\kappa)\big|_{\lambda=0,\alpha=0}}{2}\,. \quad (4.102)$$

Factorisation becomes rapidly inefficient for operators $X$ of length $\ell(X) \simeq 10$ or larger as the number of terms in the factorised form of the correlation functions grows very rapidly. In particular, it was so far inefficient for the calculation of the asymptotics of multi-point correlation functions. Yet, for these other methods, based e.g. on effective field theoretical descriptions [40] or on the use of form factor series [41], are available.

It is an interesting open question, if factorisation of static correlation functions is a peculiarity of the XXZ quantum spin chain or rather a generic feature of integrable lattice models.

## 4.3 Properties of the generalised density matrix

While the derivation of the multiple-integral formula and the construction of the Fermionic basis are rather technical and certainly exceed the scope of these lecture notes, a characterisation of the reduced density matrix by its properties is comparatively easy.

(i) The 'normalisation condition',

$$\mathrm{tr}_{1,\dots,m}\big\{D^{(N)}(\xi_1,\dots,\xi_m|T,\kappa,\kappa')\big\} = 1\,, \quad (4.103)$$

   is obvious from the definition (4.91).

(ii) The 'exchange relation',

$$\check{R}_{j-1,j}(\xi_{j-1},\xi_j)D^{(N)}(\xi_1,\dots,\xi_m|T,\kappa,\kappa')$$
$$= D^{(N)}(\xi_1,\dots,\xi_{j-2},\xi_j,\xi_{j-1},\xi_{j+1},\dots,\xi_m|T,\kappa,\kappa')\check{R}_{j-1,j}(\xi_{j-1},\xi_j)\,, \quad (4.104)$$

   where $\check{R}(\lambda,\mu) = PR(\lambda,\mu)$, is a consequence of the Yang-Baxter equation and can most easily be seen using its graphical form (2.58).

(iii) The 'left and right reduction properties',

$$\mathrm{tr}_m\big\{D^{(N)}(\xi_1,\dots,\xi_m|T,\kappa,\kappa')\big\} = D^{(N)}(\xi_1,\dots,\xi_{m-1}|T,\kappa,\kappa')\big\}\,, \quad (4.105\mathrm{a})$$
$$\mathrm{tr}_1\big\{\Theta_1(\alpha)D^{(N)}(\xi_1,\dots,\xi_m|T,\kappa,\kappa')\big\} =$$
$$\rho(\xi_1|\kappa,\kappa')D^{(N)}(\xi_2,\dots,\xi_m|T,\kappa,\kappa')\,, \quad (4.105\mathrm{b})$$

   are again obvious from (4.90) and (4.91).

These properties hold for any fundamental model (with $U(1)$-symmetry). Further properties follow from special properties of the $R$-matrix, as e.g. group invariance, asymptotics as a function of the spectral parameter, crossing symmetry.

**Remark.** The left reduction fixes the one-point functions connected with the $U(1)$-symmetry. Setting $m = 1$ and taking the derivative of (4.105b) with respect to $\kappa'$ at $\kappa' = \kappa$ we obtain

$$\mathrm{tr}_1\big\{\hat{\varphi}_1 D^{(N)}(\xi_1|T, \kappa, \kappa)\big\} = T\partial_\kappa \ln\big(\Lambda_0(\xi_1|\kappa)\big). \tag{4.106}$$

Sending $N \to \infty$, setting $\xi_1 = 0$ and using (3.76) we conclude that

$$\langle\hat{\varphi}_1\rangle = -\partial_\kappa f(T, \kappa) \tag{4.107}$$

which is consistent with the standard thermodynamic relation.

## 4.4   Comments

Using the graphical notation it is not difficult to explain the reduced qKZ equation which has be used in order to calculate the reduced density matrix of the XXZ chain. We define

$$Y_{1,\ldots,n+1} = \qquad\qquad\qquad\qquad\qquad \tag{4.108}$$

Here we suppose that there are arbitrarily many horizontal lines and the spectral parameter on the lowest line is $u$. We further assume that $|\Psi\rangle$ is a transfer matrix eigenstate. Then

$$\mathrm{tr}_n Y_{1,\ldots,n+1}\big|_{\xi_n=\xi_{n+1}=u} = \mathrm{tr}_{n+1} Y_{1,\ldots,n+1}\big|_{\xi_n=\xi_{n+1}=u} \tag{4.109}$$

by construction. It follows that

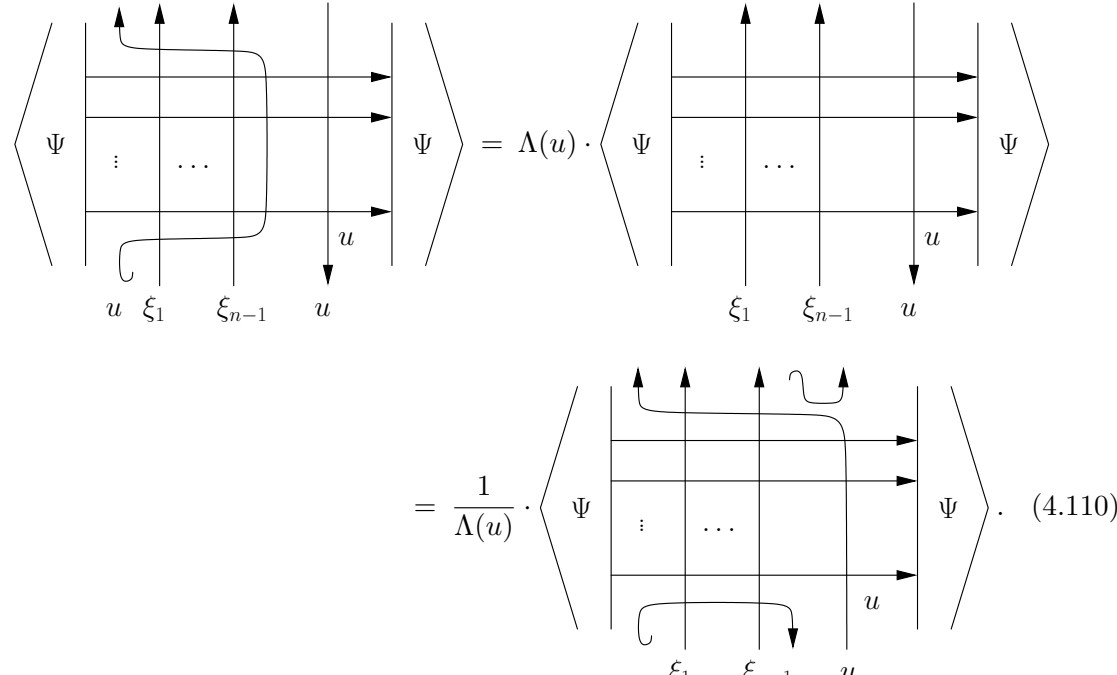

$$\tag{4.110}$$

If the $R$-matrix exhibits crossing symmetry, like in case of the XXZ chain, then the arrow direction of the rightmost transfer matrix on the right hand side of the first equation can be reversed and the second equation can be interpreted as a discrete version [23] of the reduced qKZ equation [35].

# L5   Bethe Ansatz and nonlinear integral equation for the quantum transfer matrix of the XXZ chain

## 5.1   Algebraic Bethe Ansatz for the quantum transfer matrix

In this section we consider the staggered and twisted inhomogeneous monodromy matrix (2.45), where $R$ is the $R$-matrix (2.50) of the XXZ spin-$\frac{1}{2}$ chain. Then the auxiliary space '$a$' is two-dimensional and $T_a$ can be interpreted as a $2 \times 2$ matrix with operator-valued entries acting on $\mathbb{C}^{2 \otimes N}$,

$$T_a(\lambda|\alpha) = \begin{pmatrix} A(\lambda) & B(\lambda) \\ C(\lambda) & D(\lambda) \end{pmatrix}_a . \tag{5.111}$$

The Yang-Baxter algebra relations (2.47) are a set of quadratic relations for these entries. These relations allow one to construct a set of eigenvectors of the quantum transfer matrix

$$t(\lambda|\alpha) = A(\lambda) + D(\lambda) \tag{5.112}$$

generated over a pseudo vacuum $|0\rangle$ which has the properties

$$C(\lambda)|0\rangle = 0 , \quad A(\lambda)|0\rangle = a(\lambda)|0\rangle , \quad D(\lambda)|0\rangle = d(\lambda)|0\rangle \tag{5.113}$$

for some complex functions $a(\lambda)$, $d(\lambda)$.

The existence of a pseudo vacuum is a non-trivial requirement. There are representations of the Yang-Baxter algebra which do not have a pseudo vacuum. For the quantum transfer matrix of the XXZ chain, however, a pseudo vacuum does exist. This can be easily inferred from the structure of the $R$-matrices composing the staggered monodromy matrix (2.45). They take the form

$$R_{a,j}(\lambda, \nu) = \begin{pmatrix} e_j^1 + b(\lambda - \nu)e_j^2 & c(\lambda - \nu)e_j^1 \\ c(\lambda - \nu)e_j^2 & b(\lambda - \nu)e_j^1 + e_j^2 \end{pmatrix}_a , \tag{5.114a}$$

$$R^{t_1}{}_{j,a}(\nu, \lambda) = \begin{pmatrix} e_j^1 + b(\nu - \lambda)e_j^2 & c(\nu - \lambda)e_j^2 \\ c(\nu - \lambda)e_j^1 & b(\nu - \lambda)e_j^1 + e_j^2 \end{pmatrix}_a . \tag{5.114b}$$

Setting

$$|0\rangle = \left(e_1 \otimes e_2\right)^{\otimes N/2} \tag{5.115}$$

we see that

$$T_a(\lambda|\alpha)|0\rangle = \begin{pmatrix} a(\lambda) & B(\lambda) \\ 0 & d(\lambda) \end{pmatrix}_a |0\rangle , \tag{5.116}$$

where

$$a(\lambda) = \mathrm{e}^{\alpha/2} \prod_{j=1}^{N/2} b(\nu_{2j} - \lambda) , \quad d(\lambda) = \mathrm{e}^{-\alpha/2} \prod_{j=1}^{N/2} b(\lambda - \nu_{2j-1}) . \tag{5.117}$$

For the density matrix of the grand canonical ensemble we set as before $\nu_{2j-1} = h_R/(NT)$, $\nu_{2j} = -h_R/(NT)$, $j = 1, \ldots, N/2$, and $\alpha = \kappa/T$.

For any set $\{\lambda\} = \{\lambda_j\}_{j=1}^M \subset \mathbb{C}$ we define

$$Q(\lambda|\{\lambda\}) = \prod_{j=1}^M \mathrm{sh}(\lambda - \lambda_j)\,, \tag{5.118}$$

$$|\{\lambda\}\rangle = B(\lambda_M)\dots B(\lambda_1)|0\rangle\,. \tag{5.119}$$

Then we have the following

**Theorem.** *Algebraic Bethe Ansatz [42].*

$$t(\lambda|\alpha)|\{\lambda\}\rangle = \Lambda(\lambda|\{\lambda\})|\{\lambda\}\rangle \tag{5.120}$$

*with*

$$\Lambda(\lambda|\{\lambda\}) = \frac{a(\lambda)Q(\lambda - \eta|\{\lambda\}) + d(\lambda)Q(\lambda + \eta|\{\lambda\})}{Q(\lambda|\{\lambda\})} \tag{5.121}$$

*if* $\{\lambda\}$ *is chosen in such a way that the 'Bethe Ansatz equations'*

$$\frac{d(\lambda_j)Q(\lambda_j + \eta|\{\lambda\})}{a(\lambda_j)Q(\lambda_j - \eta|\{\lambda\})} = -1\,, \tag{5.122}$$

$j = 1,\dots,M$, *are satisfied.*

**Remark.** All eigenstates are of Bethe Ansatz form (5.119), (5.122) and form a basis, if $\alpha$, $\nu_j$, $j = 1,\dots,M$, are generic [43].

## 5.2 Auxiliary functions

We may assume that $\alpha$ and the $\nu_j$ are generic. Otherwise we slightly change their values to make them generic. Then all eigenstates and eigenvalues of the quantum transfer matrix $t(\lambda|\alpha)$ can be labeled by solutions $\{\lambda_j^{(n)}\}_{j=1}^{M_n}$ of the Bethe Ansatz equations (5.122). Inserting these solutions back into (5.118) we define

$$Q_n(\lambda) = \prod_{j=1}^{M_n} \mathrm{sh}(\lambda - \lambda_j^{(n)}) \tag{5.123}$$

and the auxiliary functions

$$\mathfrak{a}_n(\lambda) = \frac{d(\lambda)Q_n(\lambda + \eta)}{a(\lambda)Q_n(\lambda - \eta)}\,. \tag{5.124}$$

By construction these functions have the important property that

$$1 + \mathfrak{a}_n(\lambda_j^{(n)}) = 0\,, \quad j = 1,\dots,M_n\,. \tag{5.125}$$

## 5.3 Nonlinear integral equations

Equation (5.125) allows us to characterise the auxiliary functions by means of nonlinear integral equations. Let $\mathcal{C}_n$ be a simple closed contour that encircles $\{\lambda_j^{(n)}\}_{j=1}^{M_n}$ and the $N/2$-fold pole of $\mathfrak{a}_n$ at $-\frac{h_R}{NT}$, but no other poles or roots of $1 + \mathfrak{a}_n$. Then the following 'monodromy condition' holds,

$$\int_{\mathcal{C}_n} \frac{\mathrm{d}\lambda}{2\pi\mathrm{i}}\, \partial_\lambda \ln\big(1 + \mathfrak{a}_n(\lambda)\big) = M_n - \frac{N}{2} = -s_n\,. \tag{5.126}$$

We shall call $s_n$ the '(pseudo-) spin of the $n$th excited state'.

In order to avoid case differentiations we restrict the parameter $\eta$ from now on to $\eta = -\mathrm{i}\gamma,\ \gamma \in (0, \pi/2]$. We would, like to point out, however, that a similar analysis is possible in all physically relevant parameter regimes. Define

$$\varepsilon_0^{(N)}(\lambda) = \kappa - \frac{TN}{2} \ln\left(\frac{\operatorname{sh}(\lambda - \frac{h_R}{NT})\operatorname{sh}(\lambda + \frac{h_R}{NT} + \eta)}{\operatorname{sh}(\lambda + \frac{h_R}{NT})\operatorname{sh}(\lambda - \frac{h_R}{NT} + \eta)}\right) \tag{5.127}$$

and

$$K(\lambda) = \operatorname{cth}(\lambda - \eta) - \operatorname{cth}(\lambda + \eta). \tag{5.128}$$

Assuming that $\lambda \pm \mathrm{i}\gamma$ is outside the contour $\mathcal{C}_n$ defined above, for all $\lambda \in \mathcal{C}_n$ and for all $\mu$ on and inside $\mathcal{C}_n$ we obtain the identity

$$\int_{\mathcal{C}_n} \frac{\mathrm{d}\mu}{2\pi\mathrm{i}} \ln\left(\frac{\operatorname{sh}(\eta + \lambda - \mu)}{\operatorname{sh}(\eta - \lambda + \mu)}\right) \partial_\mu \ln\left(1 + \mathfrak{a}_n(\mu)\right)$$

$$= \sum_{j=1}^{M_n} \ln\left(\frac{\operatorname{sh}(\eta + \lambda - \lambda_j^{(n)})}{\operatorname{sh}(\eta - \lambda + \lambda_j^{(n)})}\right) - \frac{N}{2}\ln\left(\frac{\operatorname{sh}(\eta + \lambda + \frac{h_R}{NT})}{\operatorname{sh}(\eta - \lambda - \frac{h_R}{NT})}\right)$$

$$= \ln\left(\mathfrak{a}_n(\lambda)\right) + \frac{\varepsilon_0^{(N)}(\lambda)}{T} - \mathrm{i}\pi s_n$$

$$= \ln\left(\frac{\operatorname{sh}(\eta + \lambda - x_n)}{\operatorname{sh}(\eta - \lambda + x_n)}\right)\int_{\mathcal{C}_n}\frac{\mathrm{d}\mu}{2\pi\mathrm{i}}\,\partial_\mu\ln\left(1 + \mathfrak{a}_n(\mu)\right)$$

$$+ \int_{\mathcal{C}_n}\frac{\mathrm{d}\mu}{2\pi\mathrm{i}}\left(\operatorname{cth}(\lambda - \mu + \eta) - \operatorname{cth}(\lambda - \mu - \eta)\right)\ln(1 + \mathfrak{a}_n)(\mu)$$

$$= -s_n \ln\left(\frac{\operatorname{sh}(\eta + \lambda - x_n)}{\operatorname{sh}(\eta - \lambda + x_n)}\right) - \int_{\mathcal{C}_n}\frac{\mathrm{d}\mu}{2\pi\mathrm{i}}\,K(\lambda - \mu)\ln(1 + \mathfrak{a}_n)(\mu). \tag{5.129}$$

Here we have to supply several comments and explanations. Generally, some care is necessary, when we take the logarithm of a meromorphic function and even more if we integrate it up along a contour. The first logarithm under the integral on the left hand side is defined by its principal branch. Then, due to our prerequisites, it defines a holomorphic function of $\mu$ inside and on the contour $\mathcal{C}_n$ for all $\lambda \in \mathcal{C}_n$. The logarithmic derivative under the contour is meromorphic with simple poles with residue 1 at the Bethe roots and a simple pole with residue $-N/2$ at $-\frac{h_R}{NT}$. This explains the first equation. The second equation may be understood as fixing the branch in the definition of the logarithm of the function $\mathfrak{a}_n(\lambda)$. In the third equation we perform a partial integration of the integral on the left hand side of the equation. This requires that we define the function $\ln(1 + \mathfrak{a}_n)$ as a holomorphic function, having no jumps of $2\pi\mathrm{i}$, as we move along the contour $\mathcal{C}_n$. For this purpose we fix any point $x_n \in \mathcal{C}_n$ and define a contour $\mathcal{C}_{x_n}^\lambda$ running from $x_n$ to $\lambda$ in positive direction along $\mathcal{C}_n$. Then

$$\ln(1 + \mathfrak{a}_n)(\lambda) = \int_{\mathcal{C}_{x_n}^\lambda} \mathrm{d}\mu\,\partial_\mu\ln(1 + \mathfrak{a}_n(\mu)) + \ln\left(1 + \mathfrak{a}_n(x_n)\right), \tag{5.130}$$

where the rightmost logarithm is defined by its principal branch, has the required properties.

Equation (5.129) can be interpreted as nonlinear integral equation for the auxiliary function $\mathfrak{a}_n$. For this purpose we rewrite it in the form

$$
\ln\big(\mathfrak{a}_n(\lambda)\big) =
$$

$$
- s_n \ln\left(\frac{\mathrm{sh}(\lambda - x_n + \eta)}{\mathrm{sh}(\lambda - x_n - \eta)}\right) - \frac{\varepsilon_0^{(N)}(\lambda)}{T} - \int_{\mathcal{C}_n} \frac{\mathrm{d}\mu}{2\pi\mathrm{i}} \, K(\lambda - \mu) \ln(1 + \mathfrak{a}_n)(\mu). \quad (5.131)
$$

Note that the explicit dependence on $x_n$ vanishes for states with $s_n = 0$. Another possibility to simplify the appearance of equation (5.131) occurs if the contour $\mathcal{C}_n$ can be deformed in such a way that we can send $\mathrm{Re}\, x_n \to -\infty$. Then

$$
\ln\big(\mathfrak{a}_n(\lambda)\big) = 2\mathrm{i}\gamma s_n - \frac{\varepsilon_0^{(N)}(\lambda)}{T} - \int_{\mathcal{C}_n} \frac{\mathrm{d}\mu}{2\pi\mathrm{i}} \, K(\lambda - \mu) \ln(1 + \mathfrak{a}_n)(\mu). \quad (5.132)
$$

## 5.4 Back to the roots

We just argued that every solution $\{\lambda_j^{(n)}\}_{j=1}^{M_n}$ of the Bethe Ansatz equations corresponds to an auxiliary function $\mathfrak{a}_n$ subject to the monodromy condition (5.126) and solving the non-linear integral equation (5.131). This reasoning can be reversed, performing the following steps.

(i) Exponentiate (5.131). It follows that $\mathfrak{a}_n$ has an $N/2$-fold pole at $\lambda = -\frac{h_R}{NT}$ (located inside $\mathcal{C}_n$).

(ii) The monodromy condition (5.126) then implies that $1 + \mathfrak{a}_n$ has precisely $M_n = N/2 - s_n$ roots inside $\mathcal{C}_n$.

(iii) Going backwards through partial integration we see that these roots must satisfy the Bethe Ansatz equations (5.122).

Summing up, we have seen that the Bethe Ansatz equations are in one-to-one correspondence to pairs $s_n, \mathcal{C}_n$, where the $\mathcal{C}_n$ denote equivalence classes of simple closed contours.

## 5.5 Comments

(i) We did not explain the algebraic Bethe Ansatz in any detail, since this has been done elsewhere in the literature (e.g. [3, 42]) and during the Les Houches 2018 summer school on Integrability in Atomic and Condensed Matter Physics.

(ii) For the description of the thermodynamics of the spin chain only the dominant eigenvalue $\Lambda_0$ and the corresponding auxiliary function $\mathfrak{a}_0$ are needed. These will be identified in the next section.

(iii) There are other methods for deriving non-linear integral equations and representations of the eigenvalues involving their solutions, most importantly the 'method of functional equations'. For further reading we recommend [10].

# L6 Identification of the dominant state and free energy per lattice site of the XXZ chain

## 6.1 Identification of the dominant state

In order to describe the thermodynamics of the XXZ chain we have to find out which spin value $s_0$ and which contour $\mathcal{C}_0$ belong to the dominant state. The answer can be guessed by considering the special cases $\Delta = 0$, $T \to 0$, $T \to +\infty$ and by performing numerical

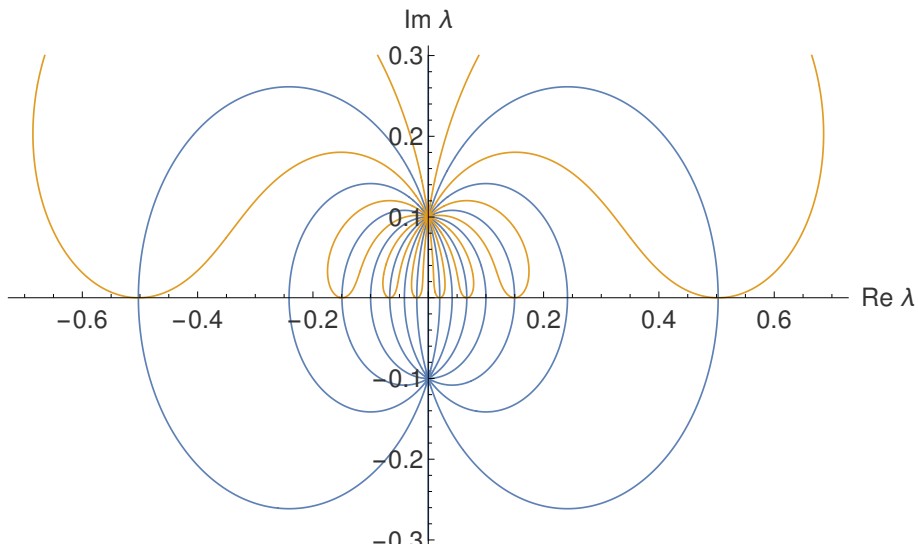

Figure 1: Contour plot of the function $f_\varepsilon$ for $N = 16$, $\varepsilon = 0.1$. Blue contours Im $f_\varepsilon = 0$, orange contours Re $f_\varepsilon = -1$. We see $8 = N/2$ points $\lambda_j$ on the real line at which $f_\varepsilon(\lambda_j) = -1$.

calculations for small Trotter numbers $N$. For space-time limitations we restrict ourselves to the consideration of the high-$T$ limit here in which we obtain a particularly clear and simple picture.

### 6.1.1 Bethe Ansatz at high temperature

Let us have a closer look at the explicit form of the auxiliary functions $\mathfrak{a}_n$ introduced in (5.124). Recalling that $h_R = -2\mathrm{i}J\sin(\gamma)$ for the XXZ chain and that we agreed upon restricting $\gamma$ to the interval $(0, \pi/2]$ for simplicity, we see that

$$\varepsilon = \frac{2J\sin(\gamma)}{NT} > 0 \tag{6.133}$$

and that this number becomes arbitrarily small for large $T$. Inserting it into the explicit expression for $\mathfrak{a}_n$ we obtain

$$\mathfrak{a}_n(\lambda) = \mathrm{e}^{-\kappa/T}\left(\frac{\mathrm{sh}(\lambda + \mathrm{i}\varepsilon)}{\mathrm{sh}(\lambda - \mathrm{i}\varepsilon)}\right)^{\frac{N}{2}}\left(\frac{\mathrm{sh}(\lambda + \mathrm{i}(\gamma - \varepsilon))}{\mathrm{sh}(\lambda - \mathrm{i}(\gamma - \varepsilon))}\right)^{\frac{N}{2}}\prod_{j=1}^{M_n}\frac{\mathrm{sh}(\lambda - \lambda_j^{(n)} - \mathrm{i}\gamma)}{\mathrm{sh}(\lambda - \lambda_j^{(n)} + \mathrm{i}\gamma)}, \tag{6.134}$$

where $\{\lambda\}_n = \{\lambda_j^{(n)}\}_{j=1}^{M_n}$ is a solution to the Bethe Ansatz equations $\mathfrak{a}_n(\lambda_j^{(n)}) = -1$, $j = 1, \ldots, M$. We would like to find 'perturbative solutions' for small $\varepsilon > 0$. One particular such solution is almost obvious. We shall see that it describes the dominant state in the high-temperature limit. This solution is 'generated' by the second factor on the right hand side of (6.134). The latter has an $N/2$-fold pole at $\mathrm{i}\varepsilon$ and an $N/2$-fold zero at $-\mathrm{i}\varepsilon$. If $\varepsilon$ is small, pole and zero are very close to each other and a model of this function for small $|\lambda|$ is

$$f_\varepsilon(\lambda) = \left(\frac{\lambda + \mathrm{i}\varepsilon}{\lambda - \mathrm{i}\varepsilon}\right)^{\frac{N}{2}}. \tag{6.135}$$

Close to the zero and close to the pole there are $N/2$ directions in which the phase of this function is $\mathrm{i}\pi$. They are connected by lines on which $f_\varepsilon$ is real negative and goes from

zero to minus infinity. Since $f_\varepsilon(\lambda)$ takes values on the unit circle for $\lambda$ on the real line, there are $N/2$ solutions $\lambda_j$, $j = 1, \ldots, N/2$, of the equation $f_\varepsilon(\lambda) = -1$ on the real axis which all go to zero for $\varepsilon \to 0$. This is sketched in figure 1. Thus, setting $M_n = N/2$ and inserting the $\lambda_j$ for $\lambda_j^{(n)}$ into the fourth factor on the right hand side of (6.134), we see that the product of third and fourth factor goes to 1 for $\varepsilon \to 0$. Since the first factor goes to 1 as well in the high-$T$ limit, we have obtained a special high-temperature solution.

### 6.1.2 A special high-temperature solution

In order to formalise this we set

$$\lambda_j = \frac{x_j}{T}\,, \tag{6.136}$$

$j = 1, \ldots, N/2$. We will look for a high-$T$ solution of the Bethe equations with $|x_j| < R$ for some $R > 0$. Setting $\lambda = x/T$, inserting (6.136) into (6.134) and sending $T \to +\infty$ the Bethe Ansatz equations turn into

$$\left(\frac{x - \frac{h_R}{N}}{x + \frac{h_R}{N}}\right)^{\frac{N}{2}} = -1\,, \tag{6.137}$$

or, equivalently,

$$p(x) = \left(x - \frac{h_R}{N}\right)^{\frac{N}{2}} + \left(x + \frac{h_R}{N}\right)^{\frac{N}{2}} = 0\,. \tag{6.138}$$

Now $p$ is a polynomial of order $N/2$ with asymptotics $p(x) \sim 2x^{N/2}$ for $x \to \infty$. Thus, there are $x_1, \ldots, x_{N/2} \in \mathbb{C}$ such that

$$p(x) = 2 \prod_{j=1}^{N/2} (x - x_j)\,. \tag{6.139}$$

### 6.1.3 The corresponding eigenvalue

The corresponding eigenvalue is

$$\Lambda(\lambda) = e^{\frac{\kappa}{2T}} \prod_{j=1}^{N/2} \frac{\operatorname{sh}(\lambda + \frac{h_R}{NT})\operatorname{sh}(\lambda - \lambda_j - \eta)}{\operatorname{sh}(\lambda - \lambda_j)\operatorname{sh}(\lambda + \frac{h_R}{NT} - \eta)} + e^{-\frac{\kappa}{2T}} \prod_{j=1}^{N/2} \frac{\operatorname{sh}(\lambda - \frac{h_R}{NT})\operatorname{sh}(\lambda - \lambda_j + \eta)}{\operatorname{sh}(\lambda - \lambda_j)\operatorname{sh}(\lambda - \frac{h_R}{NT} + \eta)}$$

$$\xrightarrow[T \to +\infty]{} \prod_{j=1}^{N/2} \frac{x + \frac{h_R}{N}}{x - x_j} + \prod_{j=1}^{N/2} \frac{x - \frac{h_R}{N}}{x - x_j} = 2\,. \tag{6.140}$$

Here we have used (6.138) and (6.139) in the last equation.

### 6.1.4 Full spectrum the in high-temperature limit

Observe that

$$t_\infty = \lim_{T \to +\infty} t(0|\alpha) = \lim_{T \to +\infty}$$

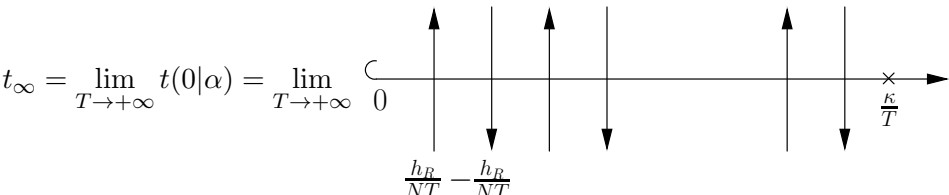

$$= |u\rangle\langle v| \quad (6.141)$$

due to the regularity (2.61) of the $R$-matrix. Clearly $|u\rangle\langle v|$ is a one-dimensional projector. Moreover,

$$\langle v|u\rangle = \bigcirc = 2 . \quad (6.142)$$

Thus, the spectrum of $t_\infty$ is $\{2, 0\}$, where the eigenvalue 0 is $2^N - 1$-fold degenerate. This means that the dominant state in the high-temperature limit is non-degenerate and has eigenvalue the $\Lambda(0) = 2$. Comparing with (6.140) above we see that the corresponding Bethe roots are $\lambda_j = x_j/T$, $j = 1, \ldots, N/2$, where the $x_j$ are the roots of the polynomial $p$.

### 6.1.5 Bethe roots of the dominant state in the Trotter limit

It is not difficult to calculate these roots explicitly. For this purpose we have to solve the Bethe Ansatz equations in the high-temperature limit,

$$\left(\frac{x_j - \frac{h_R}{N}}{x_j + \frac{h_R}{N}}\right)^{\frac{N}{2}} = -1 , \quad (6.143)$$

$j = 1, \ldots, N/2$. Clearly, if $x_j$ is a root, then $-x_j$ is a root, and if $N/2$ is odd, then $x_j = 0$ is a root. Taking the logarithm of (6.143) and setting

$$\text{tg}\left(\frac{\varphi_j}{2}\right) = \frac{2J\sin(\gamma)}{Nx_j} \quad (6.144)$$

we obtain, for any non-zero root $x_j$,

$$\frac{N}{2}\ln\left(\frac{1 + \frac{\text{i}2J\sin(\gamma)}{Nx_j}}{1 - \frac{\text{i}2J\sin(\gamma)}{Nx_j}}\right) = \frac{\text{i}N\varphi_j}{2} = \text{i}(2j - 1)\pi . \quad (6.145)$$

Using once more (6.144) and solving for $x_j$ we arrive at

$$x_j = \frac{2J\sin(\gamma)}{N\,\text{tg}\left(\frac{(2j-1)\pi}{N}\right)} , \quad (6.146)$$

where, due to the $\pi$-periodicity of the tangent function, we may restrict the range of $j$ to $-N/4 + 1 \le j \le N/4$ if $N/2$ is even or $-N/4 + 3/2 \le j \le N/4 - 1/2$ if $N/2$ is odd. In the Trotter limit, $N \to +\infty$, the roots get confined in the interval $(2J\sin(\gamma)/\pi) \times [-1, 1]$ and accumulate at the origin. The outer roots converge to $\frac{2J\sin(\gamma)}{(2j-1)\pi}$. This behaviour is illustrated with an example in figure 2.

Exercise: Find the other roots of the equation

$$\mathfrak{a}(\lambda) = \text{e}^{-\kappa/T}\prod_{j=1}^{N/2}\frac{\text{sh}(\lambda - \frac{h_R}{NT})\,\text{sh}(\lambda + \frac{h_R}{NT} - \eta)}{\text{sh}(\lambda + \frac{h_R}{NT})\,\text{sh}(\lambda - \frac{h_R}{NT} + \eta)}\frac{\text{sh}(\lambda - x_j/T + \eta)}{\text{sh}(\lambda - x_j/T - \eta)} = -1 . \quad (6.147)$$

Answer: For $T \to +\infty$ we have $N/2$ roots close to $+\eta$ and $N/2$ roots close to $-\eta$, which can be seen by setting $\lambda = z/T \pm \eta$ and sending $T \to +\infty$.

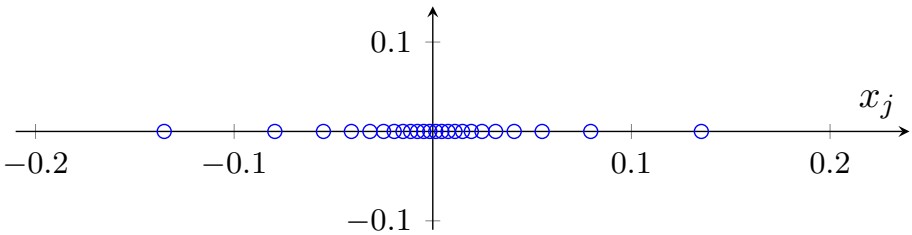

Figure 2:  Example for a configuration of the roots $x_j$ for $N = 52$, $J = 1$, $\gamma = 0.7$.

### 6.1.6  Dominant state contour

For the dominant state we may thus choose the contour

$$\mathcal{C}_0 = [-R - \mathrm{i}\gamma_-/2, R - \mathrm{i}\gamma_-/2] \cup [R - \mathrm{i}\gamma_-/2, R + \mathrm{i}\gamma_-/2]$$
$$\cup [R + \mathrm{i}\gamma_-/2, -R + \mathrm{i}\gamma_-/2] \cup [-R + \mathrm{i}\gamma_-/2, -R - \mathrm{i}\gamma_-/2] \quad (6.148)$$

with $R > 0$ large enough. As required, $\mathcal{C}_0$ encloses all Bethe roots of the dominant state in the high-temperature limit, but no other root of the equation $\mathfrak{a}_0(\lambda) = -1$ and no pole of this function other than the pole at $\lambda = -\frac{h_R}{NT}$. Since $s = 0$ for the dominant state, as seen above, we conclude that the auxiliary function of the dominant state satisfies the nonlinear integral equation

$$\ln\big(\mathfrak{a}_0(\lambda)\big) = -\frac{\varepsilon_0^{(N)}(\lambda)}{T} - \int_{\mathcal{C}_0} \frac{\mathrm{d}\mu}{2\pi\mathrm{i}} \, K(\lambda - \mu) \ln(1 + \mathfrak{a}_0)(\mu) \qquad (6.149)$$

with $\mathcal{C}_0$ according to (6.148).

## 6.2  Trotter limit and free energy per lattice site

### 6.2.1  Trotter limit

As follows from 6.1.5 the Bethe roots stay confined close to $\lambda = 0$ for $N \to +\infty$. We therefore obtain the auxiliary function in the Trotter limit by replacing $\varepsilon_0^{(N)}$ by its limit

$$\varepsilon_0(\lambda) = \lim_{N\to\infty} \varepsilon_0^{(N)} = \kappa - 2\mathrm{i}J\sin(\gamma)\,\mathrm{e}(\lambda), \quad \mathrm{e}(\lambda) = \mathrm{cth}(\lambda) - \mathrm{cth}(\lambda + \eta) \qquad (6.150)$$

in the nonlinear integral equation (6.149), resulting in

$$\ln\big(\mathfrak{a}_0(\lambda)\big) = -\frac{\varepsilon_0(\lambda)}{T} - \int_{\mathcal{C}_0} \frac{\mathrm{d}\mu}{2\pi\mathrm{i}} \, K(\lambda - \mu) \ln(1 + \mathfrak{a}_0)(\mu) \,. \qquad (6.151)$$

### 6.2.2  Free energy per lattice site

Consider the integral

$$\frac{\kappa}{2T} + \int_{\mathcal{C}_0} \frac{\mathrm{d}\mu}{2\pi\mathrm{i}} \, \mathrm{e}(\mu - \lambda) \ln(1 + \mathfrak{a}_0)(\mu)$$

$$= \ln\big(1 + \mathfrak{a}_0(\lambda)\big) + \frac{\kappa}{2T}$$
$$+ \int_{\mathcal{C}_0'} \frac{\mathrm{d}\mu}{2\pi\mathrm{i}} \left( \partial_\mu \ln\left( \frac{\mathrm{sh}(\mu - \lambda)}{\mathrm{sh}(\mu - \lambda + \eta)} \right) \right) \ln(1 + \mathfrak{a}_0)(\mu) \mod 2\pi\mathrm{i}$$

$$= \ln\big(1 + \mathfrak{a}_0(\lambda)\big) + \frac{\kappa}{2T}$$

$$- \int_{\mathcal{C}_0'} \frac{\mathrm{d}\mu}{2\pi\mathrm{i}} \, \ln\left(\frac{\mathrm{sh}(\mu - \lambda)}{\mathrm{sh}(\mu - \lambda + \eta)}\right) \partial_\mu \ln(1 + \mathfrak{a}_0)(\mu) \quad \mathrm{mod} \; 2\pi\mathrm{i}$$

$$= \ln\big(1 + \mathfrak{a}_0(\lambda)\big) + \frac{\kappa}{2T} + \ln\left(\frac{Q_0(\lambda - \eta)}{Q_0(\lambda)}\right) + \frac{N}{2} \ln\left(\frac{\mathrm{sh}(\lambda + \frac{h_R}{NT})}{\mathrm{sh}(\lambda + \frac{h_R}{NT} - \eta)}\right) \quad \mathrm{mod} \; 2\pi\mathrm{i}$$

$$= \ln\big(\Lambda_0(\lambda)\big) \quad \mathrm{mod} \; 2\pi\mathrm{i}. \tag{6.152}$$

Here $\mathcal{C}_0'$ is a modification of the contour $\mathcal{C}_0$ such that $\mathcal{C}_0 - \mathcal{C}_0'$ is a small positively oriented circle around $\lambda$. In the partial integration in the second equation we have used that $s = 0$, implying that there are no boundary terms. Equation (6.152) determines the eigenvalue in the Trotter limit.

Recalling (3.76) we obtain the free energy per lattice site of the XXZ chain in the thermodynamic limit,

$$f(T, h) = -\frac{\kappa}{2} - T \int_{\mathcal{C}_0} \frac{\mathrm{d}\lambda}{2\pi\mathrm{i}} \, \mathrm{e}(\lambda) \ln(1 + \mathfrak{a}_0)(\lambda) \tag{6.153}$$

where $\mathfrak{a}_0$ is the solution of the nonlinear integral equation (6.151).

For the identification of the dominant state and the corresponding auxiliary function we have considered the high-temperature limit here. This brought us to the conclusion that $s_0 = 0$ and that a possible contour $\mathcal{C}_0$ is the contour defined in (6.148). There are many good reasons to believe that (6.153) and (6.151) with the same choice of the contour hold for all $T > 0$.

## 6.3   Comments

(i) As we mentioned in the introduction, the latter claim is supported by numerical studies at finite Trotter number (for a pedagogical review see [44]), by a low-temperature analysis (see e.g. [15]) and by considering the XX chain (this is recommended as an exercise, for some information see [45]). In addition we would like to recommend the work [12], where the case of high but finite temperature was treated with full mathematical rigour.

(ii) The high-temperature analysis presented for the dominant state can be extended to obtain a large class of excited states in the high-temperature limit. We shall only sketch the calculation and leave the details as an exercise. Let us look for a solution $\{\lambda_j\}_{j=1}^M$ of the Bethe Ansatz equations (5.122) that has the following high-temperature asymptotics:

$$\lim_{T \to +\infty} \lambda_j \neq 0 \quad \text{for } j = 1, \ldots, n, \tag{6.154a}$$

$$\lambda_j \sim \frac{x_j}{T} \quad \text{with } |x_j| < R \text{ for some } R > 0 \text{ for } j = n + 1, \ldots, M. \tag{6.154b}$$

Inserting this high-temperature Ansatz into the Bethe Ansatz equation (5.122) and performing the limit $T \to +\infty$, we see that the first $n$ equations decouple and become

$$\left(\frac{\mathrm{sh}(\lambda_j + \mathrm{i}\gamma)}{\mathrm{sh}(\lambda_j - \mathrm{i}\gamma)}\right)^{s+n} \prod_{k=1}^n \frac{\mathrm{sh}(\lambda_j - \lambda_k - \mathrm{i}\gamma)}{\mathrm{sh}(\lambda_j - \lambda_k + \mathrm{i}\gamma)} = -1, \tag{6.155}$$

for $j = 1, \ldots, n$. These equations can be interpreted as a set of so-called higher-level equations for the high-$T$ limit. They resemble the Bethe Ansatz equations of the spin-1 XXZ chain.

Taking the product over all $j = 1, \ldots, n$ in (6.155) we obtain the 'momentum quantisation condition'

$$
\left( \prod_{j=1}^{n} \frac{\mathrm{sh}(\lambda_j + \mathrm{i}\gamma)}{\mathrm{sh}(\lambda_j - \mathrm{i}\gamma)} \right)^{s+n} = 1 \quad \Leftrightarrow \quad \prod_{j=1}^{n} \frac{\mathrm{sh}(\lambda_j + \mathrm{i}\gamma)}{\mathrm{sh}(\lambda_j - \mathrm{i}\gamma)} = \mathrm{e}^{\frac{2\pi\mathrm{i}\ell}{s+n}} \tag{6.156}
$$

for some $\ell \in \{0, 1, \ldots, s + n - 1\}$ that depends on $\{\lambda_j\}_{j=1}^{n}$. Inserting the $\lambda_j$, $j = n+1, \ldots, M$, into the Bethe Ansatz equations (5.122), performing the limit $T \to +\infty$ and using (6.156), we obtain a set of equations that determine the $x_j$,

$$
\left( \frac{x_j - \frac{h_R}{N}}{x_j + \frac{h_R}{N}} \right)^{\frac{N}{2}} = (-1)^{s+n-1} \, \mathrm{e}^{-\frac{2\pi\mathrm{i}\ell}{s+n}} \ . \tag{6.157}
$$

Depending on $N, \ell, s, n$ this equation may admit a root $x_j = 0$. All other roots are given by

$$
x_j = \frac{2J \sin(\gamma)}{N \, \mathrm{tg} \left( \frac{2\pi}{N} \left( k(j) - \frac{\ell}{s+n} \right) \right)} \ , \tag{6.158}
$$

where $k(j)$ is integer, if $s + n$ is even, or half-odd integer, if $s + n$ is odd. This means that we may choose the $M - n$ roots $x_j$ from a set of $N/2$ inequivalent values, giving $\binom{N/2}{M-n}$ different solutions.

In [12] the high-$T$ limit was worked out on more rigorous grounds, starting from the nonlinear integral equations for the excited states.

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
