# Peer review of "Statistical mechanics of integrable quantum spin systems"

_SciPost Physics Lecture Notes_

## Round 1 · Referee Report · Anonymous (Referee 1) · 2020-5-10

Strengths

1- Well written introductory text, good summary.

Weaknesses

no significant weaknesses

Report

This is a well written introductory material. It is supposed to be a Lecture Notes. The author was asked to explain some part of the literature where he contributed significantly with original research works.

I only have minor comments.

Requested changes

  1. Short typo: ''an inaccuracies''
  2. When the density matrix is introduced, I find that paragraph too quick and short. Perhaps the author could mention, that if the chain is coupled with the environment, and the whole system is in a ''pure state'', then the correlations of the chain can be described by the density matrix. Perhaps this is what the author meant with mentioning experiments, but it is useful to spell this out.
  3. When the GGE is mentioned the ref [8] is cited. This is certainly an important paper, but I would not give only this as further reading to students, because it is a very brief 4 page paper for PRL. I would add the reviews https://arxiv.org/abs/1603.06452 https://arxiv.org/abs/1604.03990 or anything similar.
  4. In 2.45 I think the partial transpose is not explained.
  5. The proof of the RTT relations for the QTM is very a difficult for a student who sees this first. I don't think that this should be left as an exercise. Perhaps some comments or instructions, or references to other lecture notes should be given.
  6. Typo: ''has be used''

  • validity: top
  • significance: high
  • originality: ok
  • clarity: top
  • formatting: perfect
  • grammar: perfect

Author:  Frank Göhmann  on 2020-06-06  [id 850]

(in reply to Report 2 on 2020-05-10)

Thank you for your positive and helpful report. I have changed the manuscript according to your suggestion. Just point 4 confused me as the partial transposed is explained right after 2.45 in equation (2.46).

---

## Round 1 · Referee Report · Anonymous (Referee 2) · 2020-5-15

Strengths

  1. Pedagogical quality of the lecture course
  2. Good overview of the subject

Weaknesses

None

Report

This is an excellent lecture course perfectly corresponding to the ``les Hoches series'' requirement. It gives rather large overview of the subject while remaining quite pedagogical and accessible for graduate students in statistical mechanics and quantum field theory. The lectures are well written and in my opinion should be published after some minor additions and corrections lister bellow.

Requested changes

  1. I think there is a misprint in the introduction (end of page 4) the word quantum (in quantum transfer matrix approach) should be rather inside the quotation marks than outside.
  2. In the beginning of the subsection 1.2.2 the number of lattice sites is denoted $N$ this notation can lead to a confusion with the Trotter number denoted also by $N$ throughout the lectures.
  3. Misprint in the eq. 1.22 (the density matrix is non-negatively definite)
  4. In the remark to the section 4.1 the paper J. M. Maillet and V. Terras, Nucl. Phys. B 575, 627 (2000) should be cited together with [16] and [17]
  5. There is a misprint in the first phrase of the section 4.4, should be probably has to be used?.
  6. In the section 5.1 the original paper L. D. Faddeev, E. K. Sklyanin and L. A. Takhtajan, Theor. Math. Phys. 40, 688 (1979) should be cited for the Algebraic Bethe Ansatz
  7. For the eqs. (5.129), (6.148) and (6.152) illustrations of the integration contours can be useful for a reader
  8. More details of the computations in (6.152) can be useful (like it was done for example for the computation in (5.129)).

  • validity: top
  • significance: high
  • originality: -
  • clarity: top
  • formatting: excellent
  • grammar: excellent

Author:  Frank Göhmann  on 2020-06-06  [id 849]

(in reply to Report 1 on 2020-05-15)

I would like to thank the referees for their constructive criticism. I have included all suggested references and tried to comply with all other suggestions. I did not take up point 2 and point 8 of the first report. I think that there is very little danger of confusion with the letter "N" in subsection 1.2.2, since it comes well before the Trotter number is introduced. And for point 8 I simply did not know what to add. In my eyes the presentation is already rather detailed.

---

## Editorial Decision

resubmitted